# Engineering a conserved RNA regulatory protein repurposes its biological function *in vivo*

Vandita D Bhat[1†], Kathleen L McCann[2†], Yeming Wang[2], Dallas R Fonseca[3], Tarjani Shukla[1], Jacqueline C Alexander[3], Chen Qiu[2], Marv Wickens[4], Te-Wen Lo[5], Traci M Tanaka Hall[6]*, Zachary T Campbell[1]*

[1]Department of Biological Sciences, University of Texas Dallas, Richardson, United States; [2]Epigenetics and Stem Cell Biology Laboratory, National Institute of Environmental Health Sciences, National Institutes of Health, Research Triangle Park, United States; [3]Department of Biology, Ithaca College, Ithaca, United States; [4]Department of Biochemistry, University of Wisconsin-Madison, Madison, United States; [5]Department of Biology, Ithaca College, Ithaca, United States; [6]Epigenetics and Stem Cell Biology Laboratory, National Institute of Environmental Health Sciences, National Institutes of Health, Research Triangle Park, United States

*For correspondence:
hall4@niehs.nih.gov (TMTH);
zachary.campbell@utdallas.edu
(ZTC)

[†]These authors contributed
equally to this work

Competing interests: The
authors declare that no
competing interests exist.

Reviewing editor: Timothy W
Nilsen, Case Western Reserve
University, United States

**Abstract** PUF (PUmilio/FBF) RNA-binding proteins recognize distinct elements. In *C. elegans*, PUF-8 binds to an 8-nt motif and restricts proliferation in the germline. Conversely, FBF-2 recognizes a 9-nt element and promotes mitosis. To understand how motif divergence relates to biological function, we first determined a crystal structure of PUF-8. Comparison of this structure to that of FBF-2 revealed a major difference in a central repeat. We devised a modified yeast 3-hybrid screen to identify mutations that confer recognition of an 8-nt element to FBF-2. We identified several such mutants and validated structurally and biochemically their binding to 8-nt RNA elements. Using genome engineering, we generated a mutant animal with a substitution in FBF-2 that confers preferential binding to the PUF-8 element. The mutant largely rescued overproliferation in animals that spontaneously generate tumors in the absence of *puf-8*. This work highlights the critical role of motif length in the specification of biological function.
DOI: https://doi.org/10.7554/eLife.43788.001

## Introduction

Post-transcriptional control of mRNA permeates biology. RNA-binding proteins control every aspect of mRNA function including processing, localization, stability, and translational status. These factors serve pivotal roles in memory, nociception, and early development (*Crittenden et al., 2002*; *Dubnau et al., 2003*; *Barragán-Iglesias et al., 2018*). RNA-binding proteins associate with sequences and structures typically situated in untranslated regions (UTRs) of an mRNA. Understanding the specificity of proteins for their regulatory motifs is crucial as these liaisons govern mRNA fate.

RNA-binding proteins comprise 4% and 10% of the human and yeast proteomes, respectively (*Castello et al., 2012*; *Beckmann et al., 2015*). A driver of this expansion is gene duplication. For example, RNA recognition motifs (RRMs), a large RNA-binding protein family, have proliferated throughout evolution (*Dreyfuss et al., 1993*; *Ray et al., 2013*) and have diverged to acquire distinct RNA recognition properties and biological functions (*Chaudhury et al., 2010*; *Zaharieva et al., 2015*). The molecular and structural underpinnings that enable diversification of RNA recognition are fundamental, as they dictate which mRNAs are subject to regulation and are a key source of evolutionary plasticity in the configuration of mRNA regulatory networks.

PUF proteins, named for PUM (Pumilio) and FBF (*fem-3* binding factor), are an exemplary system for understanding the divergence of RNA recognition within eukaryotic RNA-binding protein families (*Neeb et al., 2017*; *Wilinski et al., 2017*). PUF proteins are present in multiple copies ranging between 1 and 26 different proteins expressed per eukaryotic organism (*Wickens et al., 2002*). Classical PUF proteins recognize single-stranded RNA sequences. Eight PUM repeats are arranged in a crescent shape with the RNA bound on the concave face of the protein (*Edwards et al., 2001*; *Wang et al., 2001*; *Wang et al., 2002*; *Wang et al., 2009b*; *Zhu et al., 2009*). Each repeat contributes a trio of amino acid residues, termed a tripartite recognition motif (TRM), that directly contact the opposing RNA base (*Wang et al., 2002*; *Campbell et al., 2014*). The TRM determines the specific RNA base recognized through a combination of edge-on and stacking interactions (*Cheong and Hall, 2006*; *Koh et al., 2011*; *Valley et al., 2012*; *Campbell et al., 2014*). However, some PUF proteins contain an additional source of specificity, a binding pocket situated within the C-terminal region that accommodates a 5′ nucleotide upstream of the core RNA recognition sequence (*Zhu et al., 2009*; *Qiu et al., 2012*).

A striking source of variation among classical PUF proteins is the length of the RNA sequence motif (*Campbell et al., 2012a*). The prototypical PUF protein, PUM1, uses its eight PUM repeats to recognize an 8-nt motif called the PUM binding element (PBE), 5′-UGUANAUA-3′ (by IUPAC naming convention N is any nucleotide, H is A or C or U, W is A or U, D is A or G or U, and R is A or G), with each PUM repeat engaging one RNA base in a 1:1 recognition pattern. In contrast, other PUF proteins recognize longer RNA sequences with their eight PUM repeats. For instance, yeast PUF proteins preferentially bind core RNA elements with lengths of 8-nt (Puf3p – UGUAHAUA), 9-nt (Puf4p – UGUAHAHUA), and 10-nt (Puf5p – UGUAWYWDUA) (*Gerber et al., 2004*; *Lapointe et al., 2015*; *Lapointe et al., 2017*). All three motifs begin with a 5′ UGUR and end with a 3′ UA dinucleotide, yet the spacing and recognition of bases between these elements varies depending on the PUF protein (*Miller et al., 2008*; *Zhu et al., 2009*; *Wilinski et al., 2015*). PUF motifs correlate with distinct biological processes. In yeast, mRNA targets with different lengths are associated with mitochondrial function (8-nt), ribosome biogenesis (9-nt), and regulation of gene expression (10 nt) (*Gerber et al., 2004*; *Wilinski et al., 2015*). This suggests post-transcriptional regulatory networks of PUF proteins are defined by a combination of recognition pattern and motif length.

To understand the relevance of motif length to biological function, we focused on two PUF proteins expressed in the germline of *C. elegans* that recognize distinct regulatory elements - FBF and PUF-8. FBF recognizes a 9-nt motif, 5′-UGURNNAUA-3′, whereas PUF-8 binds an 8-nt sequence, 5′-UGUANAUA-3′ (*Bernstein, 2005*; *Opperman et al., 2005*). One mechanistic model to account for this key change in binding element length is curvature of the RNA-binding surface (*Wang et al., 2009b*). FBF's RNA-binding surface is flatter than the 8-nt binding PUM1, and this change in curvature appears to accommodate a longer RNA motif. Other PUF proteins that bind to motifs ≥ 9 nts also have correspondingly flatter surfaces (*Miller et al., 2008* and *Wilinski et al., 2015*). PUF-8 was engineered to have 9-nt specificity by substituting a 45-aa region of FBF-2 containing portions of repeats 4 and 5. Curvature change of FBF-2 is focused in this central region, suggesting that conferring a flattened architecture to PUF-8 produced 9-nt specificity. However, in the absence of a crystal structure of PUF-8, its degree of curvature is unknown, and the curvature of the chimeric protein could not be determined. As a result, the validity of this model is difficult to assess fully. A second mechanistic model to account for differences in preferred motif length is the identity of the TRM RNA-binding residues. In crystal structures of FBF-2 bound to RNA, repeat R5 lies opposite the site where an additional nucleotide is accommodated, but its TRM does not form typical 1:1 base-stacking and edge-interacting contacts with the RNA (*Wang et al., 2009b*). Conversely, repeat R5 of the 8-nt binding PUM1 forms specific contacts between its TRMs and the base at position 4 (*Wang et al., 2002*). PUF-8 appears to use a 1:1 recognition pattern like PUM1, binding 8-nt sequences with its eight PUM repeats. The relationship of these characteristic motif lengths to biological function is not well established.

Here, we provide evidence that both PUF protein curvature and TRM interactions can specify binding element length of FBF-2 and PUF-8. We report a crystal structure of PUF-8 in complex with an 8-nt RNA element. PUF-8's curvature is similar to that of PUM1 and therefore appears to enable its binding to an 8-nt RNA. To explore the role of TRMs in RNA length specificity, we reasoned that substitutions in the TRM of FBF-2 repeat R5 might enable a metamorphic change converting its specificity from a 9-nt to an 8-nt element. Using a modified yeast 3-hybrid screen, we identified and

analyzed TRM mutants that allow FBF-2 to favor binding to an 8-nt element. Structural analysis reveals that the repeat R5 TRM mutations allow FBF-2 to bind to an 8-nt element in a 1 repeat:1 RNA base pattern. The curvature of FBF-2 is unaltered by the TRM mutations, suggesting that these mutations alter the length specificity of FBF-2 through a distinct mechanism. Using this information, we used genome engineering to test whether a variant FBF-2 with PUF-8 motif length recognition can function in place of PUF-8 in vivo. Loss of PUF-8 in *C. elegans* causes tumor formation in a sensitized genetic background (*Racher and Hansen, 2012*). We generated a strain with a variation in FBF-2 that confers recognition of an 8-nt site. In this strain, tumor formation upon loss of PUF-8 is largely reversed, indicating that the FBF-2 variant substitutes effectively for PUF-8 to suppress overproliferation in the germline. Collectively, these data highlight the critical importance of RNA target recognition, including binding element length, in determining the biological function of PUF proteins. This work has broad implications beyond PUF proteins for understanding the evolution of functional divergence and modulation of consensus binding element features in nucleic acid binding proteins.

## Results

### The curvature of PUF-8 correlates with recognition of an 8-nt motif

The *C. elegans* genome encodes nine classical PUF proteins that cluster into four phylogenetic clades (*Figure 1A*) (*Stumpf et al., 2008*). Members of the FBF clade (FBF-1 and FBF-2) recognize the 9-nt FBF binding element or FBE, 5′-UGURNNAUA-3′ (*Figure 1B*) (*Zhang et al., 1997*; *Bernstein, 2005*). The clade containing PUF-8 and PUF-9 possesses a distinct specificity, and these proteins bind to the 8-nt PBE (*Figure 1C*) (*Opperman et al., 2005*; *Nolde et al., 2007*). To clarify the molecular basis of this divergent binding specificity, we determined a 2.6 Å crystal structure of PUF-8 bound to an 8-nt PBE RNA, 5′-UGUAUAUA-3′ (*Table 1*). The overall structure of the RNA-binding domain of PUF-8 is similar to that of other classical PUF proteins with eight α-helical PUM repeats (R1 to R8) and flanking regions at both the N- and C-termini (R1′ and R8′; *Figure 1C*). The eight repeats and two flanking regions together form a crescent shape, and the 8-nt RNA target sequence binds to the concave surface.

Target RNA bases are recognized by conserved amino acid side chains from PUF-8 repeats. The RNA binds with its 5′ end near the C-terminus of the protein (*Figure 1C*). PUM repeats R8-R5 bind to the 5′-UGUA sequence, and PUM repeats R3-R1 bind to bases 6–8, the AUA-3′ sequence (*Figure 2A,B*). The 5′-UGUA and AUA-3′ base specific interactions bracket a central region in which the 5th base turns away from the concave RNA-binding surface and stacks directly with the 4th base (*Figure 2C,D*). As a result, the 5th base is not recognized by PUF-8, and PBE RNAs with any nucleotide at position 5 are bound by PUF-8 with similar affinity (*Table 2*). R362 in R5 is located at the position that typically would form stacking interactions between the 4th and 5th RNA bases, but instead the side chain is moved aside (*Figure 2C,D*). This type of recognition was observed in crystal structures of human PUM1 and PUM2 with some RNA sequences and is termed the base-omission mode (*Lu and Hall, 2011*).

The mode of RNA recognition and overall curvature of PUF-8 are reminiscent of the one repeat to one base modularity of PUM1 (*Lu and Hall, 2011*). Superposition of the structures highlights this relationship as the RMSD is 1.2 Å over 293 Cα atoms, and the inner RNA-binding helices align with similar curvature (*Figure 1C*). In contrast, FBF-2 and PUF-8 do not superimpose well over all eight PUM repeats. Aligning repeats R5-R8 illustrates that the curvature of the PUF-8 scaffold contrasts starkly with the flatter FBF-2 scaffold (*Figure 1B*). These results indicate that PUF-8 resembles PUM1 with respect to curvature and mode of RNA recognition.

### A key role for FBF-2 repeat R5 in defining RNA length selectivity

Comparing the structure of PUF-8 in complex with an 8-nt sequence to FBF-2 in complex with a 9-nt element revealed a major difference in addition to curvature. As noted above, repeat R5 of FBF-2 lies opposite the additional central nucleotides in the FBE RNA (*Figures 1B* and *2C*). In the central region of both PUF-8 and PUM1, R5 uses its TRM to directly contact the RNA base at position 4 (*Figures 2C* and *3A*). Although the R5 TRMs of PUM1, PUF-8 and FBF-2 are identical, CQ/R (by convention C and Q are the edge-on residues and R is the stacking residue), we proposed that

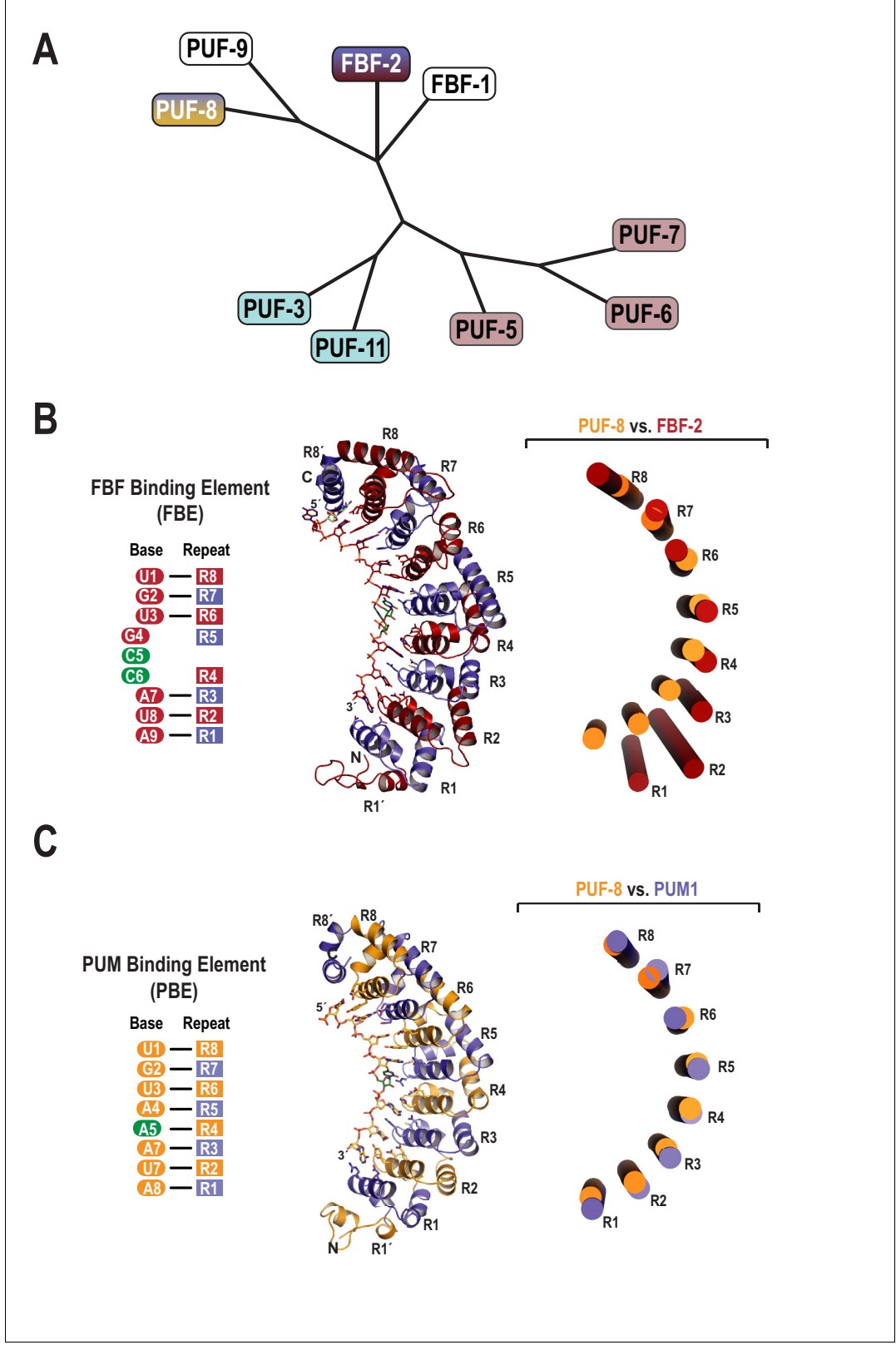

**Figure 1.** Evolutionary and structural divergence among the *C.elegans* PUF protein family. (**A**) Dendrogram of *C. elegans* PUF proteins based on alignment of primary sequences. The four clades are indicated: FBF, containing FBF-2 (blue and maroon); PUF-8/9, containing PUF-8 (blue and yellow); PUF-3/11 (cyan) and PUF-5/6/7 (mauve). (**B**) FBF-2 forms a flatter RNA-binding surface to bind to a 9-nt FBE sequence and accommodates an extra nucleotide

*Figure 1 continued on next page*

*Figure 1 continued*

opposite PUM repeats R4 and R5. Schematic illustration (left) and ribbon diagram (middle) of FBF-2 in complex with FBE RNA (PDB ID 3V74). Repeats are colored alternately red and blue. RNA recognition side chains from each PUM repeat are shown. The RNA is shown as a stick representation colored by atom type (maroon, carbon; red, oxygen; blue, nitrogen; orange, phosphorus). Carbon atoms for nucleotides 5 and 6 are green. Structural superposition of repeats R5-R8 of PUF-8 and FBF-2 demonstrates the flatter curvature of the RNA-binding surface of FBF-2. The RNA-binding helices of PUF-8 (gold) and FBF-2 (red) are shown as cylinders (right). (C) Crystal structure of PUF-8 in complex with 8-nt PBE RNA reveals modular 1:1 recognition of RNA by PUM repeats and a curvature similar to PUM1. Schematic illustration (left) and ribbon diagram (middle) of PUF-8 in complex with PBE RNA. Repeats are colored alternately gold and blue. RNA recognition side chains from each PUM repeat are shown. The RNA is colored as in panel B, except carbon atoms are gold, The RNA base at the 5$^{th}$ position, which stacks with the 4$^{th}$ base and turns away from the protein surface, is shown with green carbon atoms. Superposition of the crystal structures of PUF-8 and human PUM1 demonstrates similar curvature. The RNA-binding helices of PUF-8 (gold) and PUM1 (blue) are shown as cylinders (right).

DOI: https://doi.org/10.7554/eLife.43788.002

substitution of amino acid residues in the R5 TRM might produce interactions with the RNA that could favor binding of FBF-2 to an 8-nt sequence and offset the effects of curvature. To test this hypothesis, we developed a modified yeast three-hybrid screen to identify TRM variants that cause FBF-2 to bind preferentially to an 8-nt PBE versus a 9-nt FBE. We expressed a reporter under the control of the lac operator containing the integral membrane protein Aga2p fused to 10 HA epitope tags (*Figure 3—figure supplement 1*). In this system, the strength of an interaction drives proportional changes in the level of cell surface expression of the antigen, and we enriched for cells with high surface antigen expression using anti-HA antibodies immobilized on metal-containing resin.

We selected and validated a single FBF-2 variant both functionally and structurally (*Figure 3B,C*). The variant bears an SS/Y TRM at repeat R5, and this imparted the ability to bind to the 8-nt PBE RNA. To identify this variant, we generated a library with randomized codons in the edge-on and stacking positions of FBF-2 repeat R5. We subjected ~20,000 unique transformants to genetic and magnetic cell selection. We validated candidates using a standard yeast 3-hybrid assay where the strength of an interaction is proportional to the activity of induced β-galactosidase (*Hook et al., 2005*). We measured binding to three RNAs containing an MS2 hairpin fused to the 8-nt PBE (5′-UGUAAAUA-3′), the 9-nt FBE (5′-UGUGCCAUA-3′), or a vector sequence devoid of a known binding element (*Figure 3B*). We found that the FBF-2 variant with an SS/Y TRM preferentially bound to the PBE versus the FBE. As controls, we confirmed that FBF-2 with a wildtype R5 TRM binds preferentially to the FBE and PUF-8 binds preferentially to the PBE. We also determined the RNA-binding affinities of FBF-2 WT, FBF-2 SS/Y variant, and PUF-8 by electrophoretic mobility shift assay and confirmed preference of the FBF-2 SS/Y variant for the 8-nt PBE (*Table 2* and *Figure 3—figure supplement 2*). The binding affinity of the SS/Y variant for the PBE (22.7 nM) was comparable to the affinity of wild-type PUF-8 for the PBE (25.9 nM) and FBF-2 WT for the FBE (19.3 nM).

## Changes in curvature are dispensable

To understand how variation of the FBF-2 R5 TRM converts binding preference to an 8-nt PBE RNA, we determined a crystal structure of the R5 SS/Y variant in complex with a PBE RNA (*Table 1*) and found that the Y364 residue in the R5 TRM stacks between bases A4 and A5, which positions FBF-2's PUM repeat R5 opposite a single RNA base, A4 (*Figure 3C*). In crystal structures of wild-type FBF-2 with 9-nt RNAs, the three central RNA bases of nucleotides 4–6 form a 'triple stack' of bases that is accommodated by the flexible arginine side chain at the stacking position of repeat R5 (*Figure 3A*). In a manner similar to PUM1, Y364 of the the SS/Y variant occupies the space of the middle base of the triple stack, which allows interactions of repeats R5 and R4 with bases 4 and 5, respectively (*Figure 3A,C*). The two serine side chains (S363 and S367) contact the A4 base. S363 forms a van der Waals interaction at the C2 position, and S367 interacts via a water molecule at the N1 position. FBF-2 R5 SS/Y retained the same overall curvature in complex with an 8-nt as wildtype FBF-2 in complex with a 9-nt RNA (*Figure 3—figure supplement 3*). We conclude that the change in identity of the TRM residues is sufficient for the transformation of binding site length specificity.

**Table 1.** Data collection and refinement statistics

| Protein:RNA | | PUF-8: PBE | FBF-2 SS/Y: PBE | FBF-2 AS/Y: PBE | FBF-2 AQ/Y: PBE |
|---|---|---|---|---|---|
| **Data collection** | | | | | |
| Space group | | C2 | P61 | P61 | P61 |
| Unit Cell | $a$, $b$, $c$ (Å) | 109.2, 189.0, 63.2 | 96.4, 96.4, 99.9 | 96.5, 96.5, 101.1 | 95.9, 95.9, 100.4 |
| | $\alpha$, $\beta$, $\gamma$ (°) | 90, 103.6, 90 | 90, 90, 120 | 90, 90, 120 | 90, 90, 120 |
| Resolution (Å) | | 50–2.55 (2.59–2.55)[*] | 50–2.25 (2.29–2.25)[*] | 50–2.25 (2.33–2.25)[*] | 50–2.85 (2.9–2.85)[*] |
| $R_{sym}$ or $R_{merge}$ | | 0.191 (0.692) | 0.101 (0.704) | 0.104 (0.772) | 0.191 (0.957) |
| $I/\sigma I$ | | 9.4 (1.9) | 19.2 (3.42) | 17.1 (2.98) | 12.8 (2.34) |
| Completeness (%) | | 98.9 (98.0) | 99.9 (100) | 99.9 (100) | 99.6 (99.2) |
| Redundancy | | 6.9 (3.6) | 5.7 (5.7) | 5.7 (5.7) | 10.7 (8.8) |
| | | | | | |
| **Refinement** | | | | | |
| Resolution (Å) | | 33.8–2.6 | 32.0–2.3 | 31.6–2.3 | 27.5–2.9 |
| No. reflections | | 37,625 | 25,089 | 25,386 | 12,185 |
| $R_{work}/R_{free}$ | | 0.229/ 0.285 | 0.158/ 0.204 | 0.167/ 0.223 | 0.219/ 0.272 |
| No. atoms | | | | | |
| Protein | | 8415 | 3197 | 3194 | 3189 |
| RNA | | 507 | 150 | 168 | 168 |
| Solvent | | 229 | 169 | 109 | 21 |
| $B$-factors (Å$^2$) | | | | | |
| Wilson B | | 29.7 | 36.6 | 35.5 | 48.4 |
| Protein | | 32.8 | 45.3 | 45.1 | 50.2 |
| RNA | | 43.6 | 51.4 | 58.8 | 64.8 |
| Solvent | | 32.7 | 49.6 | 44.3 | 23.1 |
| R.m.s deviations | | | | | |
| Bond lengths (Å) | | 0.002 | 0.007 | 0.007 | 0.002 |
| Bond angles (°) | | 0.45 | 0.77 | 0.78 | 0.38 |

[*]Values in parentheses are for the highest-resolution shell.

DOI: https://doi.org/10.7554/eLife.43788.003

A key question is whether the TRM combination that favored 8-nt length preference exists in nature. To address this question, we searched a database of ~24,000 PUM repeat sequences identified in Pfam for the presence of the TRM combination capable of switching specificity (*Finn et al., 2016*). We identified the SS/Y TRM in a PUM repeat from *Naegleria gruberi* (*Figure 3D*). The observation of the SS/Y TRM in nature implies that the combination of amino acid residues reported here from in vitro selection has also arisen through natural selection.

## The stacking residue is critical for binding length specificity

To determine which amino acid residues in the R5 SS/Y TRM contribute to the transition of specificity away from the FBE and towards the PBE sequence, we generated a series of systematic substitutions to the SS/Y TRM residues. First, we addressed whether the serine residues that occupy edge-on positions are required to confer preference of the PBE. We substituted alanine for one (AS/Y or SA/Y) or both (AA/Y) of the edge-on serine residues. Intriguingly, all of the mutants with edge-on

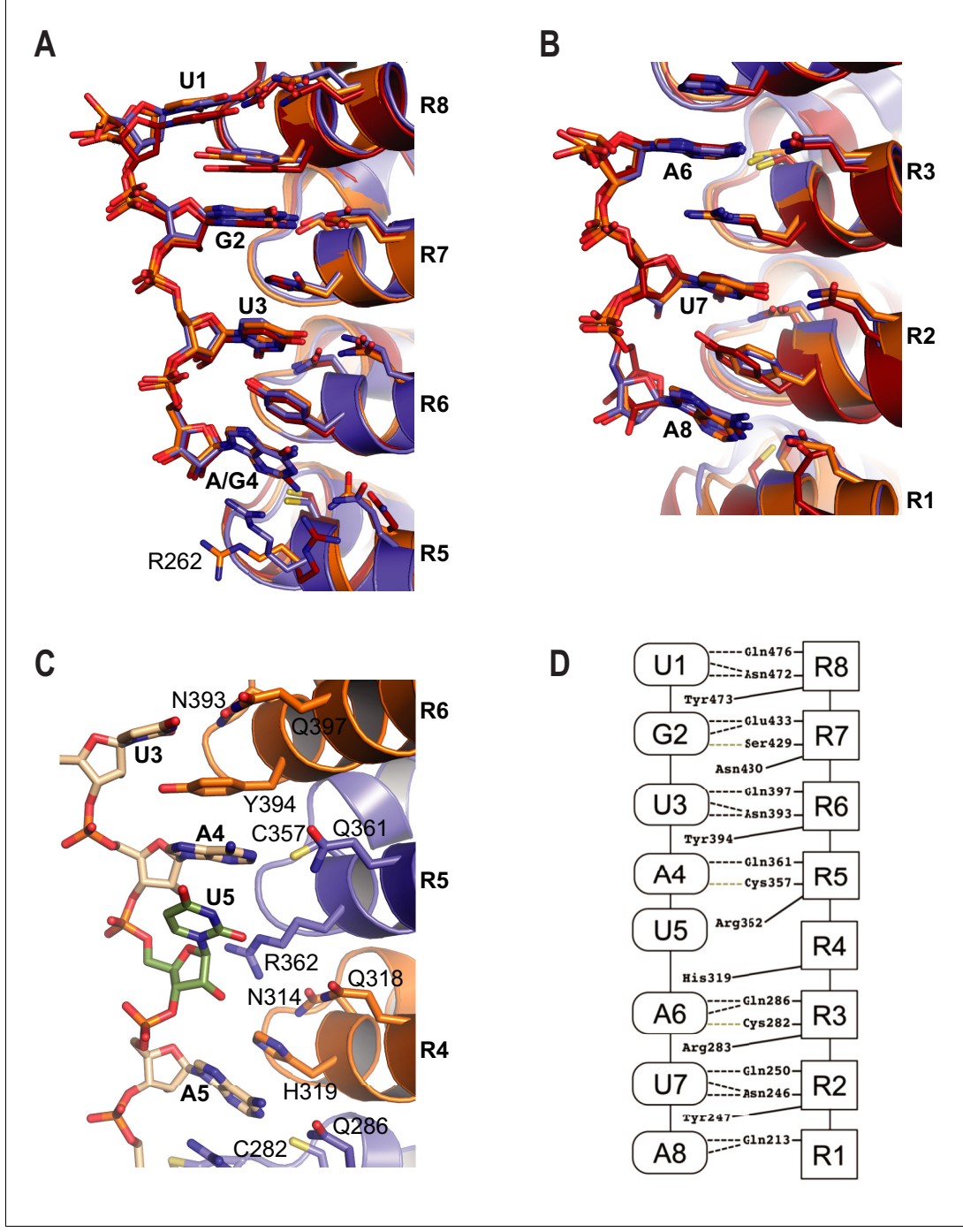

**Figure 2.** RNA recognition by PUF-8. (**A**) Recognition of the conserved 5´-UGUR sequence by PUF proteins. (**B**) Recognition of the conserved AUA-3´ sequence by PUF proteins. Superpositions of crystal structures of PUF-8:PBE RNA (orange), human PUM1:PBE RNA (blue), and *C. elegans* FBF-2/FBE RNA (red) are shown. Structures were aligned by superimposing the RNA bases. (**C**) PUF-8 base omission mode of RNA recognition. In panels A-C, the TRM residues from each PUM repeat are shown. (**D**) Schematic representation of the interactions between PUF-8 and PBE RNA. PUM repeats are indicated by boxes, and RNA bases are indicated by ovals. Interactions are indicated by dashed lines (hydrogen bonds, black; van der Waals contacts, tan).

DOI: https://doi.org/10.7554/eLife.43788.004

**Table 2.** RNA-binding analyses of PUF-8 and FBF-2 proteins

| Protein | RNA | RNA sequence | $K_d$ (nM) | $K_{rel}$[*] |
|---|---|---|---|---|
| PUF-8 | PBE | UGUA UAUA | 28.8 ± 0.7 | 1 |
| | PBE-A5 | UGUA AAUA | 25.9 ± 3.1 | 0.9 |
| | PBE-C5 | UGUA CAUA | 44.8 ± 2.4 | 1.6 |
| | PBE-G5 | UGUA GAUA | 45.6 ± 3.1 | 1.6 |
| | FBE | UGUGCCAUA | 3110 ± 656 | 108 |
| FBF-2 WT | PBE | ACAUGUAA AUAC | 74.6 ± 7.7 | 1 |
| | FBE | ACAUGUGCCAUAC | 19.3 ± 0.6 | 0.3 |
| FBF-2 SS/Y | PBE | ACAUGUAA AUAC | 22.7 ± 0.4 | 1 |
| | FBE | ACAUGUGCCAUAC | 50.9 ± 2.3 | 2.2 |
| FBF-2 AS/Y | PBE | ACAUGUAA AUAC | 16.3 ± 0.8 | 1 |
| | FBE | ACAUGUGCCAUAC | 76.7 ± 4.3 | 4.7 |
| FBF-2 SS/R | PBE | ACAUGUAA AUAC | 51.1 ± 1.1 | 1 |
| | FBE | ACAUGUGCCAUAC | 26.9 ± 2.3 | 0.5 |
| FBF-2 AQ/Y | PBE | ACAUGUAA AUAC | 20.2 ± 1.2 | 1 |
| | FBE | ACAUGUGCCAUAC | 79.9 ± 2.0 | 4.0 |

[*]$K_{rel}$ values are calculated for each protein with binding to the PBE RNA set to 1.

DOI: https://doi.org/10.7554/eLife.43788.005

The following source data is available for Table 2:

Source data 1. Data for *Table 2*.

DOI: https://doi.org/10.7554/eLife.43788.006

alanine substitutions and a tyrosine stacking residue retained preferential binding to the 8-nt PBE over the 9-nt FBE (*Figure 4A*, *Figure 4—figure supplement 1*). A crystal structure of the AS/Y variant in complex with PBE RNA was nearly identical to the SS/Y structure (*Table 1* and *Figures 3C* and *4B*). We also tested the binding preferences of an FBF-2 R5 AQ/Y variant that we identified as an in vivo substitution by genome engineering (see below). This variant demonstrated preferential binding to the 8-nt PBE over the 9-nt FBE, and a crystal structure confirms recognition of the 8-nt motif (*Figure 4A,C*, *Figure 4—figure supplement 1*). These data suggest that the tyrosine stacking residue plays a prominent role in preferential binding of FBF-2 variants to the PBE.

We next tested whether the tyrosine stacking residue is required for PBE specificity of the SS/Y variant and found that it is necessary to convert FBF-2 to 8-nt specificity. We substituted the stacking residue with alanine, which reverted FBF-2 R5 SS/A to 9-nt FBE specificity, indicating the crucial role of Y364 (*Figure 4A*). When we restored the WT stacking residue, R364, the resulting R5 SS/R variant poorly discriminated between the FBE and PBE. We sought to determine whether a tyrosine stacking residue was sufficient to convert FBF-2 to 8-nt specificity. However, a CQ/Y variant, which maintains WT edge-interacting residues, failed to bind either RNA (*Figure 4A*). Although the CQ/Y protein was expressed (*Figure 4—figure supplement 2*), it lost both PBE and FBE binding activity. Collectively, our data suggest that mutation of the stacking residue from arginine to tyrosine is necessary to direct specificity towards the PBE but small residues in the edge-on positions are also required to completely switch specificity.

## Compensatory mutations reveal engagement of the 3´ end

Our crystal structures of the FBF-2 variants indicated that they used their TRMs to recognize the full 8-nt PRE. Among PUF proteins, mutations to both the RNA and TRM residues that contact the 5´ portion of the FBE sequences tend to be less tolerated than those to the 3´ end and protein partners can enable degeneracy on the 3´ end (*Campbell et al., 2012a*; *Valley et al., 2012*; *Weidmann et al., 2016*). The interactions of the FBF-2 variants with the 3´ site could have been favored under the high concentrations of crystallization resulting in an artifactual 1:1 binding mode.

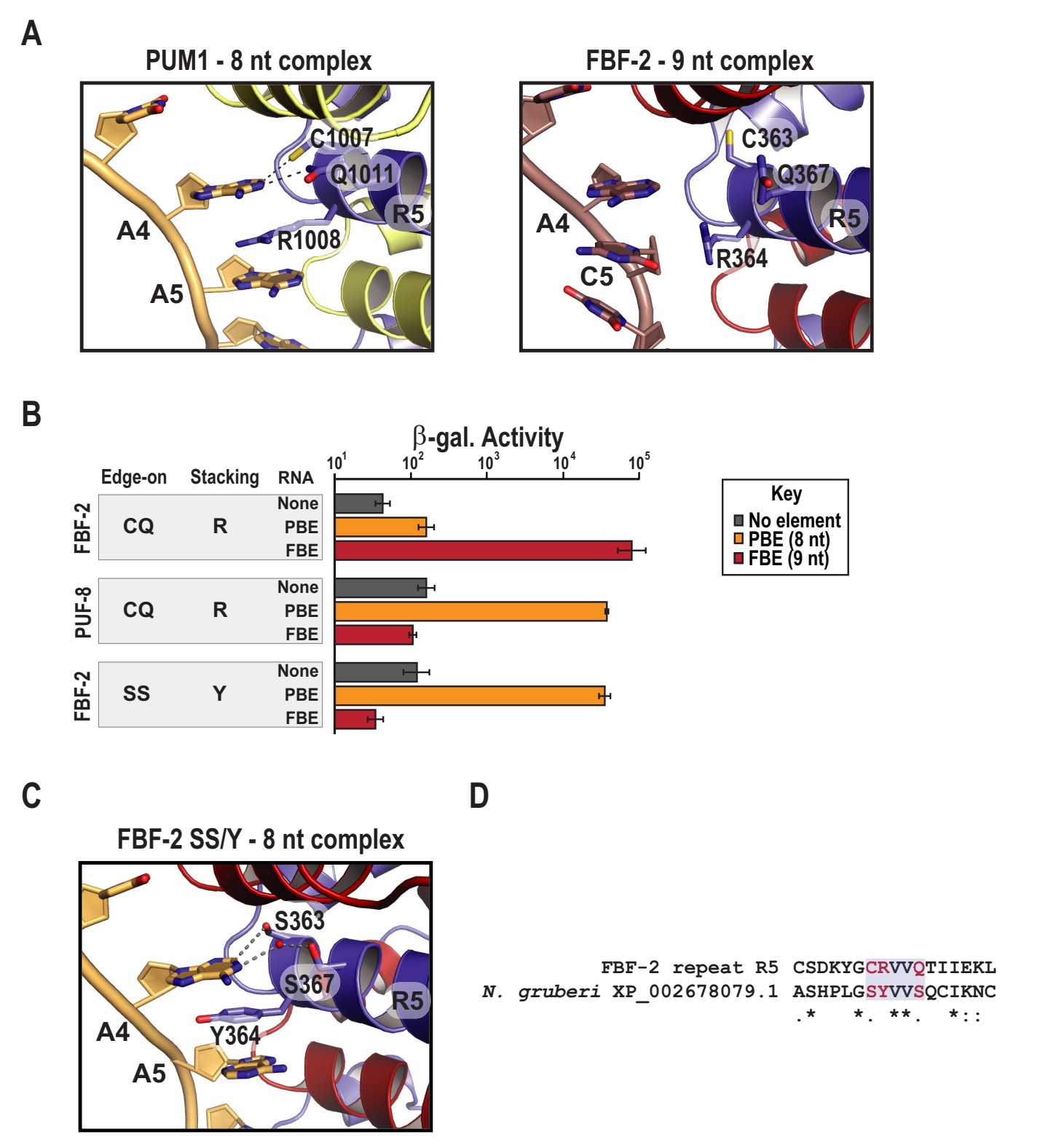

**Figure 3.** Substitution of TRM residues in FBF-2 repeat R5 to SS/Y switches specificity from a 9-nt FBE to an 8-nt PBE. (**A**) PUM1 binds preferentially to an 8-nt PBE by intercalating R1008 between bases A4 and A5 (left, PDB ID 3Q0L). This is distinct from FBF-2 bound to a 9-nt FBE where R364 projects away from base C5 (right, PDB ID 3K5Q). The PBE (gold) and FBE (mauve) RNAs are shown with cartoon backbones and stick bases. Hydrogen bond and van der Waals interactions between TRM residues and RNA bases are indicated with dashes. (**B**) An FBF-2 variant bearing the SS/Y TRM at repeat

*Figure 3 continued on next page*

*Figure 3 continued*

R5 preferentially binds to an 8-nt PBE. Yeast 3-hybrid analyses of binding by FBF-2 WT, PUF-8, and FBF-2 SS/Y variant to an MS2 hairpin (None, grey) or an MS2 hairpin fused to an 8-nt PBE (orange) or a 9-nt FBE (red). Binding activity is shown as units of β-galactosidase activity normalized to cell count. Error bars indicate the standard deviation of three replicate measurements. Source data are available in *Figure 3—source data 1*. (C) The crystal structure of the FBF-2 SS/Y variant reveals binding to the 8-nt PBE in a 1:1 recognition pattern. Hydrogen bond and van der Waals interactions are indicated with dashes, and a water molecule is shown as a red sphere. (D) Identification of a naturally occurring SS/Y TRM. Sequence alignment of FBF-2 repeat R5 with PUM repeat R7 from *N. gruberi*. TRM residues are indicated in red. Identical residues are labeled with an asterisk, dots indicate similar types of amino acid residues.

DOI: https://doi.org/10.7554/eLife.43788.007

The following source data and figure supplements are available for figure 3:

**Source data 1.** Raw data for Beta-Glo assay of FBF-2 R5 SS/Y variant with different binding elements.

DOI: https://doi.org/10.7554/eLife.43788.011

**Figure supplement 1.** Modified yeast three-hybrid system to detect RNA-protein interactions.

DOI: https://doi.org/10.7554/eLife.43788.008

**Figure supplement 2.** Electrophoretic mobility shift assays confirm preferential recognition of the PBE by SS/Y.

DOI: https://doi.org/10.7554/eLife.43788.009

**Figure supplement 3.** The FBF-2 R5 SS/Y variant binds to an 8-nt PBE without changing the overall curvature.

DOI: https://doi.org/10.7554/eLife.43788.010

Additionally, the crystal structure of the SS/Y variant bound to the PRE showed poor density for the $8^{th}$ nucleotide. To test the 1:1 binding mode, we examined whether interactions of FBF-2 variants with the 3′ sequences of the 8-nt RNA are required for tight binding in cells. The remarkable modularity of individual PUM repeats enables the generation of variants that selectively associate with RNAs containing G at the opposing position in an RNA target (*Opperman et al., 2005*; *Cheong and Hall, 2006*; *Campbell et al., 2014*; *Porter et al., 2015*). We introduced a G-selective TRM combination SE/H into repeat R2 of the FBF-2 R5 variants SS/Y, AS/Y, and AQ/Y (FBF-2 R5 SS/Y R2 SE/H, R5 AS/Y R2 SE/H, and R5 AQ/Y R2 SE/H) (*Wang et al., 2009a*; *Campbell et al., 2014*). As a control using a PUF protein that naturally binds an 8-nt sequence, we also generated a PUF-8 variant with an SE/H TRM in repeat R2 (PUF-8 R2 SE/H). We assessed the ability of these TRM variants to interact with RNAs of length from 7 to 10 nts that ended with GA at the 3′ end to match the TRMs of repeats R2 and R1, 5′-UGUA-A$_{1\text{-}4}$-GA-3′ (*Figure 4D*). If interaction with the 3′ sequences is important for binding, the repeat R2 variants should bind preferentially to the 8-nt site. In contrast, if a 5′UGUAA sequence is sufficient for tight binding, the repeat R2 variants should bind equally well to sites of all lengths. We found that the PUF-8 variant R2 SE/H and the FBF-2 variants R5 SS/Y R2 SE/H, R5 AS/Y R2 SE/H, and R5 AQ/Y R2 SE/H bound preferentially to the 8-nt U7G RNA with an order of magnitude greater activity than to the shorter and longer elements. All of the variants bound poorly to the wild-type PBE sequence due to the mismatch of a G-specific TRM in repeat R2 opposite a U7 nucleotide. We therefore conclude that the FBF-2 R5 TRM variants retain association to the 3′ end of target RNAs.

## FBF-2 R5 TRM variants retain FBF-2 base recognition specificity at positions 3–5

We assessed whether the FBF-2 R5 TRM variants that favor binding to an 8-nt PBE have base recognition properties more similar to the original scaffold, WT FBF-2, or to PUF-8 (*Figure 5*). FBF-2 and PUF-8 both specify a U3 and have loosened sequence requirements at position 5 (*Campbell et al., 2012a*). In contrast, PUF-8 is specific for A4 while FBF-2 accepts A4 or G4 (*Bernstein, 2005*). These preferences are reflected in their known target mRNAs. We tested binding of WT PUF-8 and FBF-2 R5 TRM variants SS/Y, AS/Y, and AQ/Y to 8-nt RNA sequences with base substitutions at positions 3–5 of the PBE using the yeast three-hybrid system. The FBF-2 variants R5 SS/Y, AS/Y, and AQ/Y bound the PBE with comparable activity to PUF-8 and retained specificity for a U3 base. This is consistent with prior studies that showed TRM substitutions did not affect specificity of the preceding base (*Bernstein, 2005*; *Valley et al., 2012*). At nucleotide 4, which is opposite repeat R5, PUF-8 was selective for A, as expected. However, the FBF-2 R5 TRM variants were not as selective at this position. FBF-2 R5 SS/Y bound equally well to either an A or G at position 4, similar to WT FBF-2 (*Bernstein, 2005*), and the R5 AS/Y variant accommodated A4, G4 or U4. The R5 AQ/Y variant

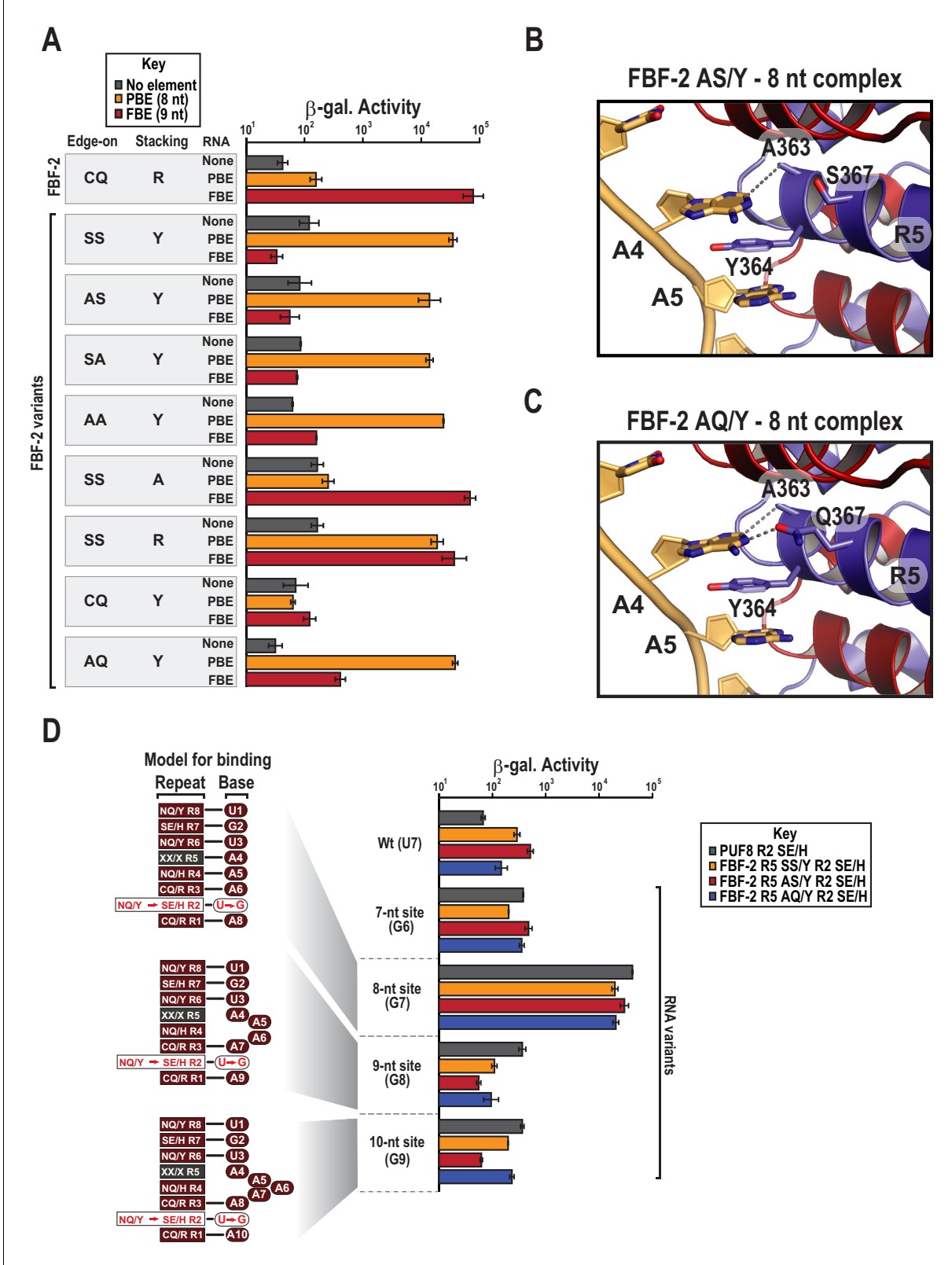

**Figure 4.** Y364 in the FBF-2 SS/Y variant is critical for 8-nt PBE selectivity. (**A**) Interaction of FBF-2 TRM variants with 8-nt PBE and 9-nt FBE RNAs. Yeast 3-hybrid analyses of binding by FBF-2 WT and the FBF-2 SS/Y variant to an MS2 hairpin (None, grey) or an MS2 hairpin fused to an 8-nt PBE (orange) or a 9-nt FBE (red). Binding activity is shown as units of β-galactosidase activity. Source data are available in *Figure 4—source data 1*. (**B**) The FBF-2 AS/Y variant binds to the PBE RNA in a 1:1 recognition pattern similar to the SS/Y variant. Hydrogen bond and van der Waals interactions are indicated with

*Figure 4 continued on next page*

*Figure 4 continued*

dashes. (C) The FBF-2 AQ/Y variant binds to the PBE RNA in a 1:1 recognition pattern. (D) The FBF-2 variants retain recognition of the 3′ sequence. Yeast 3-hybrid analyses of binding by PUF-8 and FBF-2 variants to an MS2 hairpin fused to an 8-nt WT PBE or a 7–10-nt PBE with the penultimate nucleotide changed to G. Binding activity is shown as units of β-galactosidase activity normalized to cell count. Error bars indicate the standard deviation of three replicate measurements. Mutants in PUF-8, FBF-2 SS/Y, FBF-2 AS/Y, or FBF-2 AQ/Y introduce a requirement for a G base opposite repeat R2, and interaction with only 8-nt sequences indicates the importance of the 3′ sequence. Source data areavailable in *Figure 4—source data 2*.
DOI: https://doi.org/10.7554/eLife.43788.012

The following source data and figure supplements are available for figure 4:

**Source data 1.** Raw data for Beta-Glo assay FBF-2 R5 variants with different binding elements.
DOI: https://doi.org/10.7554/eLife.43788.015
**Source data 2.** Raw data for Beta-Glo assay FBF-2 R5 variants with different binding elements.
DOI: https://doi.org/10.7554/eLife.43788.016
**Figure supplement 1.** Electrophoretic mobility shift assays.
DOI: https://doi.org/10.7554/eLife.43788.013
**Figure supplement 2.** The FBF-2 R5 CQ/Y mutant is expressed in yeast.
DOI: https://doi.org/10.7554/eLife.43788.014

displayed preference for A4, although not as strictly as PUF-8. These results are consistent with the interactions observed in the crystal structures of the FBF-2 variants with PBE RNA. The tyrosine base-stacking interaction is critical for binding energy, whereas serine or alanine residues in the edge-interacting positions permit binding to different nucleotides at position 4. The AQ/Y variant is most similar to the WT TRM, which seems to impart preference for A4. Finally, at nucleotide 5, all proteins preferred A5 and excluded G5. PUF-8 had broadened specificity and also accommodated C5 or U5, and FBF-2 R5 AS/Y and AQ/Y also bound to U5. Collectively, these experiments demonstrate that the FBF-2 R5 TRM variants retain specificity for U3 and the ability of WT FBF-2 to recognize A4 or G4. However, the TRM variation at repeat R5 has differing effects on the specificity of repeat R4 for the nucleotide at position 5. This suggests that there is cooperativity between TRMs as opposed to true independent modularity. Similar results were obtained with variants at the R7 TRM (*Campbell et al., 2014*). Finally, the FBF-2 R5 AQ/Y variant demonstrates a recognition pattern qualitatively more similar to PUF-8 than FBF-2.

## An 8-nt-binding FBF-2 variant can substitute for PUF-8 in vivo

With the knowledge of variant combinations that convert the binding preference of FBF-2 to mirror that of the homologue PUF-8, we next sought to determine whether altering the RNA recognition motif length would allow FBF-2 to fulfill PUF-8 protein function to control germline cell proliferation. Using genome engineering, we tested whether an 8-nt-binding FBF-2 variant could rescue loss of *puf-8* function in vivo. In the *C. elegans* germline, loss of *puf-8* does not substantially impair mitotic proliferation or entry into meiosis (*Subramaniam and Seydoux, 2003*; *Bachorik and Kimble, 2005*). This is in contrast with temperature-sensitive *germline proliferation-1* (*glp-1*) gain-of-function mutants where mitotic cells overproliferate (e.g. the *glp-1(ar202)* strain), resulting in a tumorous phenotype at the restrictive temperature (25°C) (*Pepper et al., 2003*). This phenotype is strongly enhanced by loss of *puf-8* as evidenced by tumors at the permissive temperature (15°C) (*Racher and Hansen, 2012*). Upon reduction of *puf-8* by RNAi (*puf-8(RNAi)*), the *glp-1(ar202)* mutant strain develops tumors throughout the germline at permissive temperature (15°C) (*Figure 6A*). To mark cells in mitosis, we used immunofluorescence to detect phosphorylated histone H3 (PHH3) in formaldehyde-fixed gonads. We also stained germline cells with DAPI to visualize nuclear morphology. Tumorous germlines were defined as having DAPI cells with anti-PHH3 staining throughout the distal and proximal end of the germline. Based on these criteria, 98% of the *glp-1(ar202) puf-8(RNAi)* germlines contained tumors despite expression of WT FBF (*Table 3*). Tumor formation in the *glp-1(ar202)* animals depended upon depletion of *puf-8*, since treatment of animals with an RNAi vector containing a scrambled RNA sequence did not result in tumorigenesis (*Table 3*).

Although PUF-8 and FBF have opposing effects on mitosis, we reasoned that both proteins are likely translational repressors and that regulatory differences may arise through changes in motif length recognition that direct their functions to distinct subsets of the transcriptome (*Friend et al., 2012*; *Vaid et al., 2013*). To test this idea, we generated a *C. elegans* strain expressing an 8-nt-

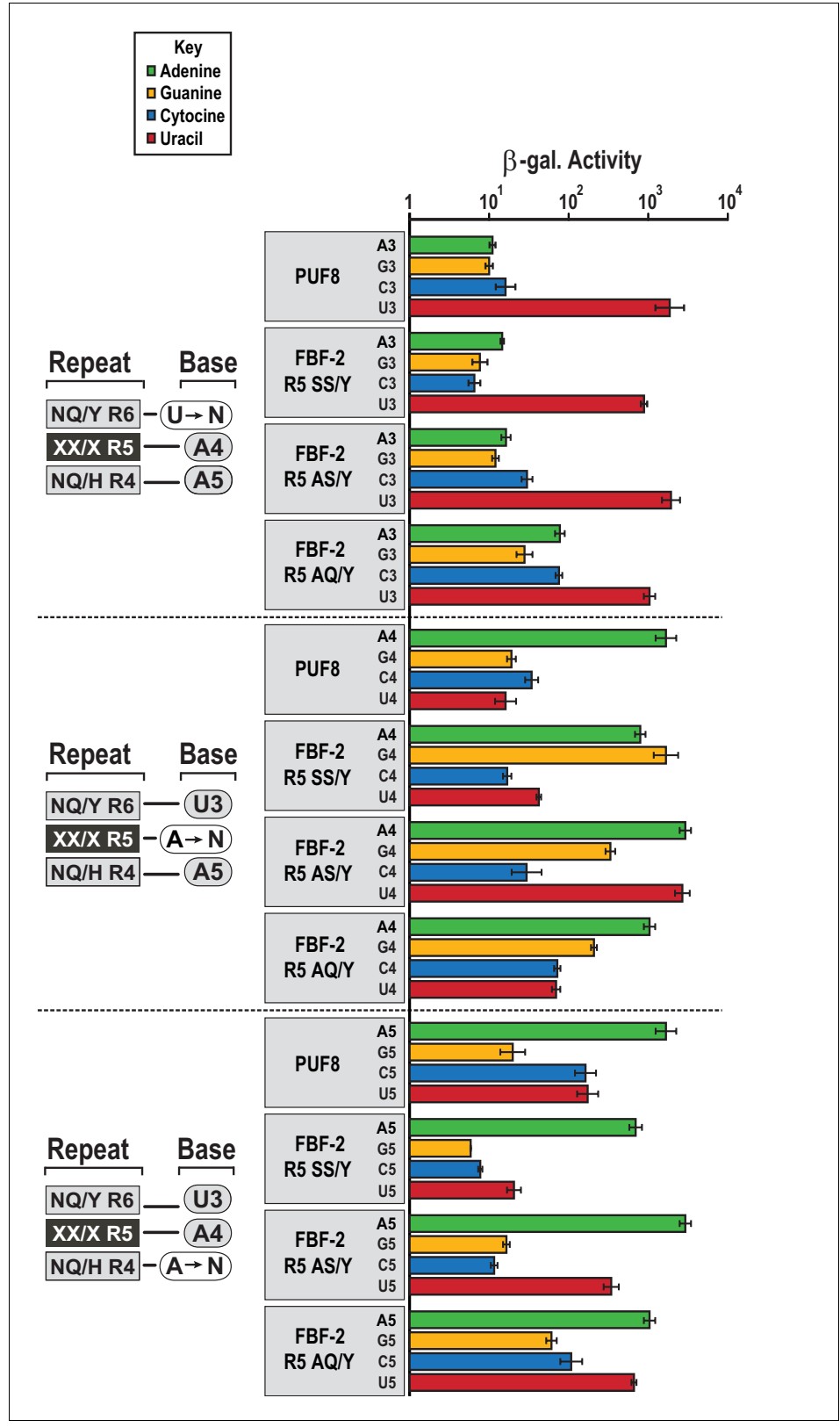

**Figure 5.** FBF-2 variants retain base recognition specificity at flanking positions. Yeast 3-hybrid analyses of binding by PUF-8, FBF-2 SS/Y, FBF-2 AS/Y, and FBF-2 AQ/Y to an MS2 hairpin fused to 8-nt PBE RNAs bearing nucleotide substitutions at positions 3–5. Binding activity is shown as units of β-galactosidase activity normalized to cell count. *Figure 5 continued on next page*

*Figure 5 continued*

Error bars indicate the standard deviation of three replicate measurements. Source data areavailable in *Figure 5— source data 1*.

DOI: https://doi.org/10.7554/eLife.43788.017

The following source data is available for figure 5:

**Source data 1.** Raw data for Beta-Glo assay of FBF-2, FBF-2 R5 variants and PUF-8 that carry R2 SE/H mutations with different length binding elements.

DOI: https://doi.org/10.7554/eLife.43788.018

binding FBF-2 variant using CRISPR. Mutations were introduced into the endogenous gene encoding FBF-2 (note that FBF-2 can be modified, because FBF-1 is functionally redundant to FBF-2). We injected Cas9 protein, a guide RNA (target sequence: 5′-AGATTTGTTCTGATAAGTAT-3′), and repair templates corresponding to the AS/Y and SS/Y mutations into the germline of N2 animals. After numerous attempts, we were unable to recover the desired AS/Y and SS/Y mutants. We did, however, recover multiple strains with deletion mutations and a single strain that incorporated a novel repeat R5 TRM variant - AQ/Y. This mutant strain was designated *fbf-2(lot14)*. We speculate that this variant resulted from homology repair when using the AS/Y repair template. Serendipitously, as shown above, the AQ/Y mutation preferentially binds to the PBE relative to the FBE in cells and in vitro (*Figure 4*, *Figure 4—figure supplement 1*, *Table 2*), and a crystal structure of the FBF-2 R5 AQ/Y variant bound to the PBE adopts a similar conformation to SS/Y and AS/Y bound to the PBE (*Figure 4C*). Furthermore, the FBF-2 R5 AQ/Y mutant retains base recognition specificity at flanking positions (*Figure 5*). We conclude that the AQ/Y variant behaves similarly to SS/Y and AS/Y and preferentially favors binding to the 8-nt PBE.

To determine if the FBF-2 R5 AQ/Y variant regulated PUF-8 targets, we generated homozygous *fbf-2(lot14)/glp-1(ar202)* double mutant animals and examined germline proliferation following depletion of PUF-8 by RNAi. We found that the edited strain expressing the FBF-2 AQ/Y variant (*fbf-2(lot-14), glp-1(ar202)* animals) displayed tumor formation in only 36% of gonads at 15°C (*Figure 6B*, *Table 3*). In contrast, tumors were formed in 98% of gonads from *glp-1(ar202)* animals (*Figure 6A*, *Table 3*). Thus, the FBF-2 AQ/Y variant is capable of substituting for PUF-8 in vivo.

## Discussion

Our experiments reveal three insights into RNA-PUF protein interactions. First, PUF-8 adopts a structure reminiscent of human PUM1 and forms base-specific contacts to seven of the eight nucleotides that comprise the binding element. Second, mutations in repeat 5 of FBF-2 can introduce new contacts to the RNA that switch preferential binding to an 8-nt PBE sequence motif over a 9-nt FBE motif. The amino acid residue combinations we obtained through random mutagenesis occur in nature. Third and finally, a variation in FBF-2 that enables it to recognize the consensus binding element of PUF-8 rescues a germline tumor phenotype caused by loss of function of *puf-8* in a sensitized background.

Prior results suggest that the degree of curvature of the RNA-binding surface of PUF proteins corresponds to the length of the RNA element that is recognized, but our results here indicate that the identities of the TRMs also contribute to the RNA motif length of PUF proteins. In the case of changing specificity of FBF-2 to that of PUF-8, we have demonstrated that a minimum of two amino acid changes (CQ/R to AQ/Y) can swap binding element length preference, but they do not impact curvature. This finding is consistent with the notion that changes in curvature are remarkably well correlated with, but not necessarily required for alterations in binding element length preference. Crystal structures of different PUF proteins in complex with their RNA targets indicate that PUF proteins with flatter RNA-binding surfaces recognize longer RNA elements (*Wang et al., 2001*; *Miller et al., 2008*; *Wang et al., 2009b*; *Wilinski et al., 2015*). In a few instances, crystal structures have been determined with and without RNA, and the degree of curvature of these PUF proteins does not change upon RNA binding (*Edwards et al., 2001*; *Wang et al., 2001*, *Wang et al., 2002*, *Miller et al., 2008*; *Weidmann et al., 2016*). Chimeras that transfer FBF-2 binding specificity to PUF-8 by substituting central segments of FBF-2 have also argued in favor of curvature as a major

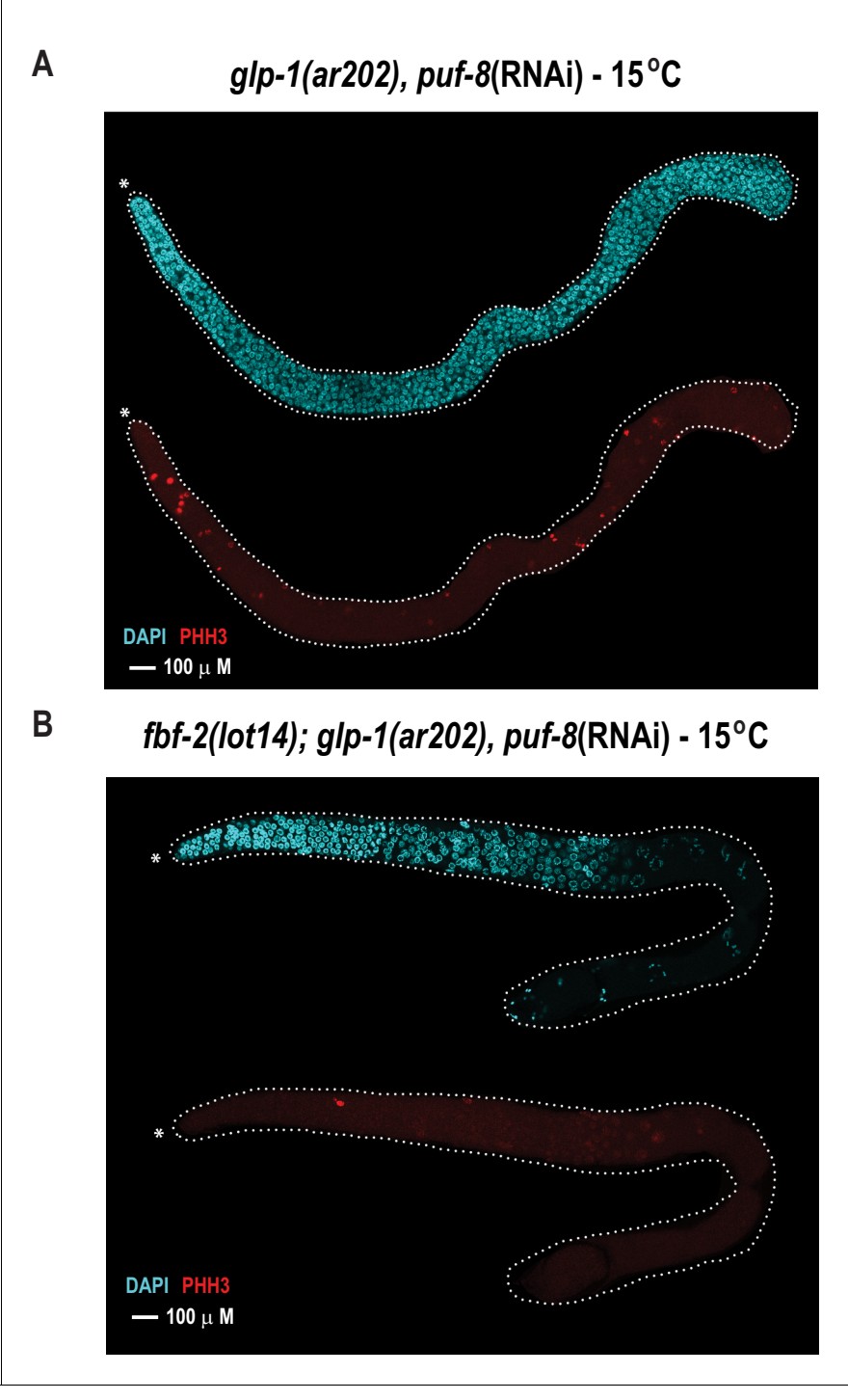

**Figure 6.** The FBF-2 R5 AQ/Y variant partially rescues the tumorous phenotype in the *C.elegans* germ line caused by loss of PUF-8 in a sensitized genetic background. Extruded germlines were stained for nuclei (DAPI, blue) or mitotic cells (α-PHH3, red). (**A**) Fluorescence microscopic image of an extruded germline from an animal with wild-type *fbf-2* and a gain-of-function mutation (*glp-1(ar202)*) that was subjected to puf-8 depletion by RNAi. These animals produce tumors throughout the germline as evidenced by the presence of red mitotic cells. (**B**) Fluorescence microscopic image of an extruded germline from an edited animal with FBF-2 AQ/Y variant (*fbf-2(lot14)*) and a gain-of-function mutation (*glp-1(gf)*) that was subjected to puf-8 depletion by RNAi. Note the absence of red mitotic cells throughout the gonad.
DOI: https://doi.org/10.7554/eLife.43788.019

**Table 3.** Phenotypic analysis of mutant strains

| Genotype | RNAi | Wild-type Germline, % | Complete tumorous Germline, % | N |
|---|---|---|---|---|
| glp-1(ar202) | puf-8 | 2 | 98 | 61 |
| glp-1(ar202) | Scramble | 100 | 0 | 95 |
| fbf-2(lot14) glp-1(ar202) | puf-8 | 64 | 36 | 89 |
| fbf-2(lot14) glp-1(ar202) | Scramble | 100 | 0 | 107 |

DOI: https://doi.org/10.7554/eLife.43788.020

driver of motif length (*Opperman et al., 2005*; *Wilinski et al., 2015*). Our crystal structure of PUF-8 confirms the difference in its curvature from that of FBF-2. Although the FBF-2-to-PUF-8 specificity switch could be engineered without curvature change, a PUF-8-to-FBF-2 specificity switch might require a curvature change. The degree of curvature of the RNA-binding surface of the chimeric protein is unknown, but the segments of FBF-2 that can transfer its specificity to PUF-8 occur at the region where curvature change originates. The distinct curvature among different PUF proteins in an organism raises the question as to whether topological differences provide additional functional benefits beyond binding element length specificity. Perhaps changes in the convex surface of the protein foster or diminish specific protein partnerships and this impacts RNA regulatory mechanism (*Campbell et al., 2012b*; *Menichelli et al., 2013*; *Wu et al., 2013*).

Our findings directly impact the growing field of synthetic biology applications of engineered PUF proteins with novel specificities (*Wang et al., 2013*). The specificity of individual TRMs has been established for several dozen combinations (*Campbell et al., 2014*). Two component systems that join programmable RNA recognition with optical tags or effector domains have broad applications for understanding and manipulating RNA function (*Ozawa et al., 2007*; *Tilsner et al., 2009*). For instance, covalent fusion of FBF-2 to a poly(A) polymerase prevented deadenylation of a targeted mRNA (*Cooke et al., 2011*). Conceptually similar approaches with PUM1 have been used to modify splicing, RNA stability, and localization (*Wang et al., 2009a*; *Dong et al., 2011*; *Choudhury et al., 2012*; *Abil et al., 2017*). One way to encode recognition of different length RNA elements is through engineering PUF proteins with additional or fewer PUM repeats (*Filipovska et al., 2011*; *Zhao et al., 2018*). Our work provides another potential means to reduce the length of the binding element, thus increasing the range of targeting modalities available for tailored RNA recognition by the PUF scaffold.

The finding that FBF-2 bearing mutations that alter motif length recognition can rescue knockdown of PUF-8 provides additional insight into PUF-8 and FBF-2 biological functions. PUF-8 and FBF-2 produce dissimilar cellular outcomes. PUF-8 restricts overproliferation: single *puf-8* mutants display proximal gonad tumors due to dedifferentiation of sperm (*Subramaniam and Seydoux, 2003*). In contrast, FBF-2 (together with FBF-1) promotes mitosis: double *fbf-1/2* mutants display profound developmental defects in cellular proliferation (*Crittenden et al., 2002*). Our experiment capitalized on the fact that PUF-8 and FBF-2 are both expressed in the mitotic region of the germline and recognize similar but non-identical motifs (*Campbell et al., 2012a*), and the activities of FBF-1 and FBF-2 are largely redundant, which allowed engineering of FBF-2 (*Lamont et al., 2004*). The ability of the FBF-2 AQ/Y variant to rescue PUF-8 function suggests that the downstream molecular consequences of PUF-8 and FBF-2 RNA recognition are similar. Their different effects on cellular proliferation are therefore directed by the functions of their respective mRNA targets.

Why modulate motif length recognition as a means of differentiating homologous RNA-binding proteins? This is a broadly conserved phenomenon in organisms with multiple PUF genes (*Wilinski et al., 2017*). Following a gene duplication event, diversification of binding specificity provides a means to regulate a unique set of targets with the potential to engender new and advantageous biological functions. One possible reason why motif length would be an attractive parameter is that the use of a conserved handle (e.g. 5′UGUA) enables evolution of the downstream sequence to sample different RNA regulons through subtle changes to the existing motif. This plasticity could enable more rapid transitions as both the RNA motif and regulators diverge and co-evolve. This principle may also apply to other RNA-binding proteins, which often comprise multiple RNA-binding modules akin to the multiple PUM repeats. For example, the RNA-binding properties of RRM

proteins expand with tandem domains. A gene duplication introducing an additional RNA-binding domain maintains recognition of the original RNA motif while specificity of the new domain evolves to produce a distinct RNA motif.

## Materials and methods

### Structural analysis

#### Protein expression and purification

PUF-8: A cDNA encoding the PUF-8 RNA-binding domain (residues T171-S525) was cloned into the vector pGEX-6P1 (GE Healthcare), which encodes an N-terminal glutathione S-transferase (GST) tag followed by a TEV protease cleavage site. The vector was transformed into *E. coli* strain BL21 star (DE3) (Invitrogen), and cultures were grown at 37°C in LB medium supplemented with 100 µg ml$^{-1}$ ampicillin until the $OD_{600}$ reached 0.6–0.8. Fusion protein expression was induced by addition of 0.3–0.5 mM isopropyl β-D-1-thiogalactopyranoside (IPTG) and incubation at 25°C for 16–20 hr.

Cell pellets were frozen at −20°C. Upon thawing, cell pellets from each liter of culture were resuspended into 25 ml lysis buffer (50 mM Tris pH 7.5, 250 mM NaCl, 250 mM $(NH_4)_2SO_4$, 5 mM β-mercaptoethanol [β-ME], 1 mM EDTA). After sonication and centrifugation of the lysate, the soluble fraction was mixed with 0.5–1.0 ml glutathione resin (Sigma) per liter of culture in a 50 ml conical tube rotating at 4°C for 3–4 hr. The mixture was then transferred into a 25 ml disposable column. The beads were washed sequentially with 100 column volumes of lysis buffer, 25 column volumes of an ATP wash buffer (20 mM Tris pH 8.5, 10 mM ATP, 5 mM $MgCl_2$), and another 100 column volumes of the lysis buffer. The GST-TEV-PUF-8 fusion protein was eluted with eight column volumes of an elution buffer (50 mM Tris pH 8.0, 50 mM NaCl, 50 mM $(NH_4)_2SO_4$, 10 mM reduced glutathione). The fusion protein was cleaved by addition of recombinant TEV protease (final concentration 10 µg ml$^{-1}$) at 4°C overnight.

To purify the PUF-8 protein from the GST tag and TEV protease, the cleaved fusion protein was diluted 2-fold with buffer A [20 mM Tris pH 8.0, 5 mM β-ME, 1 mM EDTA] and immediately loaded onto a 5 ml Hi-Trap Heparin column (GE Healthcare). PUF-8 eluted at about 18% buffer B [20 mM Tris pH 8.0, 5 mM β-ME, 1 mM EDTA, 1 M NaCl, 1 M $(NH_4)_2SO_4$]. PUF-8 protein was concentrated to ~1 mg ml$^{-1}$ and mixed with an 8-nt PBE RNA (5′-UGUAUAUA-3′) at a molar ratio of 1:1.1. The protein:RNA mixture was incubated at 4°C overnight for optimal binding and then purified by size exclusion chromatography using a Superdex 200 gel filtration column (GE Healthcare) equilibrated in a running buffer containing 50 mM Tris pH 7.5, 200 mM NaCl, 5 mM DTT, 1 mM EDTA. The purified protein:RNA complex was exchanged into a final buffer [10 mM Tris pH 7.5, 150 mM NaCl, 1 mM DTT] and concentrated to $OD_{280}$ = 5.0 (protein concentration of ~3.5 mg ml$^{-1}$) using Amicon Ultra-15 concentrators with a 10 kDa cutoff.

FBF-2: A cDNA encoding the FBF-2 RNA-binding domain (residues S164 – Q575) was cloned into the vector pSMT3 (kindly provided by Dr. Christopher Lima), which encodes an N-terminal His$_6$-SUMO fusion tag followed by a TEV protease cleavage site. The SS/Y, AS/Y, AQ/Y and SS/R mutations were introduced into FBF-2 in the pSMT3 vector using site-directed mutagenesis PCR. Nucleotide sequences for all mutants were confirmed using DNA sequencing. FBF-2 proteins were expressed in *E. coli* strain BL21 star (DE3) (Invitrogen) at 15°C for 16–20 hr in the presence of 50 µg ml$^{-1}$ kanamycin and 0.1 mM IPTG, which was added when the $OD_{600}$ reached 0.6–0.8.

Cell pellets were frozen at −80°C. Upon thawing, cell pellets from each liter of culture were resuspended in 35 ml lysis buffer (20 mM Tris pH 8, 500 mM NaCl, 20 mM imidazole pH 8, 5% (v/v) glycerol, 0.1% (v/v) β-ME). After sonication and centrifugation of the lysate, the soluble fraction was mixed with 2.5 ml Ni-NTA resin (Qiagen) per liter of culture in a 50 ml conical tube rotating at 4°C for 2 hr. The mixture was then transferred into a 50 ml disposable column. The beads were washed with 100 column volumes of lysis buffer. The His$_6$-SUMO-FBF-2 fusion proteins were eluted with 20 column volumes of elution buffer (20 mM Tris pH 8, 50 mM NaCl, 200 mM imidazole pH 8, 1 mM dithiothreitol [DTT]). The His$_6$-SUMO fusion was cleaved from FBF-2 by addition of recombinant TEV protease at 4°C overnight.

To purify the FBF-2 proteins from the His$_6$-SUMO tag and TEV protease, the cleaved fusion protein was filtered through a 0.22 µM filter and loaded onto a 5 ml Hi-Trap Heparin column (GE Healthcare) in buffer A (20 mM Tris pH 8, 1 mM DTT). FBF-2 proteins eluted at about 38–40% buffer

B (20 mM Tris pH 8, 1 M NaCl, 1 mM DTT). The fractions containing the FBF-2 proteins were pooled and concentrated to a volume <1 ml using Amicon Ultra-15 concentrators with a 30 kDa cutoff. Concentrated FBF-2 proteins purified by size selection on a HiLoad 16/60 Superdex 75 column (GE Healthcare) equilibrated in a running buffer containing 20 mM Hepes pH 7.4, 150 mM NaCl, 2 mM DTT. FBF-2 protein was concentrated to >1 mg ml$^{-1}$ using Amicon Ultra-15 concentrators with a 30 kDa cutoff.

## Crystallization

Crystals of a PUF-8:PRE RNA complex were grown by hanging drop vapor diffusion by mixing the protein:RNA complex solution at a 1:2 (v/v) ratio with crystallization solution (3.6 M sodium formate and 10 mM betaine hydrochloride or 3.6 M sodium formate and 3% [w/v] dextran sulfate). Shortly before data collection, crystals were transferred into a series of modified crystallization solutions supplemented with 5%, 10%, and finally 20% (v/v) glycerol, incubating in each solution for 5 min. Crystals were frozen by flash-cooling in liquid nitrogen.

For crystallization, FBF-2 variant proteins were mixed with PBE RNA (5′-UGUAAAUA-3′) at a molar ratio of 1:1.2 and incubated overnight at 4°C for optimal binding. Precipitated material was pelleted through centrifugation of the protein:RNA complexes at 21,000xg for 10 min, and soluble protein:RNA complexes were used for crystallization.

Crystals of FBF-2 variant complexes with PBE RNA were grown by hanging drop vapor diffusion by mixing the protein:RNA complex solution at a 1:1 (v/v) ratio with optimized crystallization solutions. The crystallization solutions were SS/Y:PBE (100 mM Tris pH 8.6, 15% [w/v] PEG 8000, 8% [v/v] ethylene glycol), AS/Y:PBE (100 mM Tris pH 8.0, 15% [w/v] PEG 8000, 8% [v/v] ethylene glycol), and AQ/Y:PBE (100 mM Tris pH 8.0, 12% [w/v] PEG 8000, 8% [v/v] ethylene glycol). Shortly before data collection, crystals were transferred into a series of modified crystallization solutions supplemented with 5%, 10%, 15%, and finally 20% (v/v) glycerol. Crystals were frozen by flash-cooling in liquid nitrogen.

## X-ray data collection and processing

Diffraction data for PUF-8 were collected from crystals at 100 K using a home X-ray source (Rigaku Micromax-007HF X-ray generator with Saturn 92 CCD detector, wavelength 1.5418 Å, NIH/NIEHS). Data were indexed and scaled with *HKL2000* (*Otwinowski and Minor, 1997*), and converted to structure factors using *SCALEPACK2MTZ* from the *CCP4i* software package (CCP4 (*Collaborative Computational Project, Number 4, 1994*). Data collection and processing statistics are shown in *Table 1*.

Diffraction data for FBF-2 variants were collected from crystals at 100 K at the SER-CAT beamline 22-ID or 22-BM at the Advanced Photon Source, Argonne National Laboratory. Data were indexed and scaled with *HKL2000* (*Otwinowski and Minor, 1997*), and converted to structure factors using *SCALEPACK2MTZ* from the *CCP4i* software package (CCP4 (*Collaborative Computational Project, Number 4, 1994*). Data collection and processing statistics are shown in *Table 1*.

## Crystal structure determination and refinement
### PUF-8

A crystal structure of the PUF-8:PBE RNA complex was determined by molecular replacement using the crystal structure of the Pumilio-homology domain of human PUM1 (PDB 1M8Y, 45% sequence identity) as the initial model (*Wang et al., 2002*). The initial structure determination was performed in 2008. *MolRep* from the *CCP4i* software package (CCP4 (*Collaborative Computational Project, Number 4, 1994*) was used to correct the initial model by alignment and find the three copies of PUF-8 in each asymmetric unit. *PHASER* from the *CCP4i* software package (CCP4 (*Collaborative Computational Project, Number 4, 1994*) was used to calculate initial phases. CNS was then used to refine the initial model at 50–2.6 Å resolution, including rigid body refinement, a simulated annealing at 5000 K to reduce model bias, grouped and individual temperature factor refinements as well as energy minimization (*Brünger and Rice, 1997*). O was used for manual rebuilding (*Jones et al., 1991*). Electron density for the bound RNA was visible in the initial electron density map, but the RNA was built at a later refinement stage when the electron density became continuous. The final model of the PUF-8:PBE RNA complex comprises residues D174 to L511 and

nucleotides U1 to A8. An N-terminal glycine residue encoded by the TEV cleavage site, N-terminal residues TTT (3-5) and C-terminal residues FQKPAVMS (518-525) were not included in the structure due to poor electron density at the N- and C-termini. *Phenix.Refine* was employed for addition of water molecules and TLS refinement (*Afonine et al., 2005*). Additional refinement and model building was performed with COOT and Phenix (Adams 2002, Emsley and Cowtan 2004). Refinement statistics are shown in *Table 1*. For each structure, all φ-ψ torsion angles are within allowed regions of the Ramachandran plot and 98% are in the most favored regions. All superpositions were calculated using *SUPERIMPOSE* from the *CCP4i* software package (*Collaborative Computational Project, Number 4, 1994*). Figures were prepared with *PyMol* (Schrödinger) (*DeLano, 2002*).

### FBF

Crystal structures of the FBF-2 mutants in complex with PBE RNA were determined by molecular replacement using the crystal structure of FBF-2:FBE (PDB ID:3K5Q) as the search model with Phaser. Iterative model building was done with COOT and Phenix (Adams 2002, Emsley and Cowtan 2004). The final models of the FBF-2:PBE RNA complexes comprise residues L168 to S569 and nucleotides U1 to A8. The density for nucleotide A8 was weak in the structure of the SS/Y mutant, so it was not modeled. Refinement statistics are shown in *Table 1*.

## Electrophoretic mobility shift assays

Equilibrium dissociation constants and percentage of active protein were determined as described previously [13]. Briefly, 100 pM of synthetic RNA (Dharmacon), $^{32}$P 5′-end-labeled using T4 polynucleotide kinase, was incubated with a range of FBF-2 protein concentrations in 10 mM Hepes (pH 7.4), 50 mM NaCl, 0.1 mg ml$^{-1}$ BSA, 0.01% (v/v) Tween-20, 0.1 mg ml$^{-1}$ yeast tRNA (Ambion), 1 mM EDTA, 1 mM DTT for 1 hr at room temperature. Loading dye (4 µl of 2.5% Ficoll 400 [v/v], 0.05% bromophenol blue) was added to each 20 µl reaction before loading 10 µl on a pre-run non-denaturing 10% TBE-polyacrylamide gel. The apparent dissociation constants were calculated using GraphPad Prism (Graphpad LLC) by fitting data from at least three independent experiments using non-linear regression with a one-site, specific binding model. The dissociation constants in *Table 2* were adjusted based on the percentage of active protein in each preparation (PUF-8, 78%; FBF-2 WT, 80%; SS/Y, 47%, SS/R, 69%, AQ/Y, 75%; AS/Y, 64%).

## Screening

A randomized library was generated through amplification of an oligonucleotide encoding R5 by PCR (template: CCGTCAGATTTGTTCTGATAAGTATGGGNNNNNNNGTTGTGNNNACTATTATCGAAAAGCTCACTGCTGA; forward primer: CCACCCCAGAGCACCTCCGTCAGATTTGTTCTGATAAGTAT; reverse primer: CAACGTTCATTGAATCAGCAGTGAGCTTTTCGATAA). The resulting product encodes randomized amino acid residues at the edge-on and stacking positions of R5 (RQICSDKYGXXVVXTIIEKLTA). A linear vector encoding FBF-2 fused to the GAL4 activation domain was generated by PCR (Forward primer: CCCATACTTATCAGAACAAATCTGACGG; Reverse primer: ACTATTATCGAAAAGCTCACTGCTGA) (*Hook et al., 2005*). The vector was combined with the insert at a 1:3 molar ratio and subjected to micro-homology guided in vitro recombination (*Gibson et al., 2009*). After library generation, screening was conducted in a modified version of YBZ-1 (pLexA Aga2p::10xHA). Yeast cells were co-transformed with plasmids encoding FBF-2 mutants and the PBE expressed from p3HR2 (*Bernstein et al., 2002*). Transformants were plated on selective media lacking histidine, uracil, tryptophan, and leucine. Colonies were re-suspended in TBST (Tris-buffered saline containing 0.05% [v/v] Tween-20) and subjected to magnetic sorting using anti-HA magnetic beads (cat. 88836, Thermo Fisher) equilibrated in TBST. After incubation at room temperature for 30 min, cells were collected by placing tubes on a magnetic stand and the supernatant was discarded. The beads were washed twice with 200 µl of TBS-T, mixed and collected by placing on a magnetic stand. The beads were re-suspended in sterile water and plated on yeast selective media agar.

## Yeast-three hybrid assays

YBZ-1 was co-transformed with pGADT7 plasmids encoding FBF-2 (residues 121–632) and PUF-8 (127–519) fused to Gal4 (*Bernstein et al., 2002*). Mutant RNAs were expressed using the p3HR2

vector. Luminescence data were collected with the β-Glo reagent (Promega) and measured using a 96-well Tecan plate reader (Tecan).

## Genome editing

N2 young adult worms were injected with the following: 3.0 µL Cas9 (61 µM) (IDT cat #: 1074181), 3.0 µL duplexed 100 µM tracrRNA (IDT cat #: 1072533), 100 µM crRNA guideRNA, (IDT cat #: 1074181), and 2.0 µL of each 100 µM AS/Y and 100 µM SS/Y repair oligos (IDT). F1 progeny were cloned into individual liquid culture wells in 96 well plates. F2 and F3 progeny were screened via PCR and using CEL1 (*Lo et al., 2013*). CEL1-positive candidates were selected for homozygotes and sequenced to identify molecular lesions. The guide RNA sequence was 5′-AGATTTGTTCTGATAAG TAT-3′. The AS/Y repair oligonucleotide sequence was 5′-cctccgtcagatttgttctgataagtatggc GCG TAT gttgtg TCC actattatcgaaaagctcactgctg-3′. The SS/Y repair oligonucleotide sequence was 5′-cctccgtcagatttgttctgataagtatggc TCA TAC gttgtg TCA actattatcgaaaagctcactgctg-3′. Strain designations were as follows: wild type N2, *glp-1(gf)*: GC833 *glp-1(ar202), and* AQ/Y: TWL013 *fbf-2(lot14)*.

### RNAi

*puf-8* RNAi vectors and feeder bacteria were obtained from Dharmacon (cat. numbers: RCE1182; ORF ID: C30G12.7). Bacteria were streaked onto selective media containing ampicillin (50 µg ml$^{-1}$) and tetracycline (12.5 µg ml$^{-1}$). Single colonies were isolated and grown in 3 ml Luria broth with ampicillin (50 µg ml$^{-1}$) at 37°C in a shaker. Cultures were concentrated prior to seeding on Nematode Growth Medium (NGM) plates. Gravid worms were synchronized by bleaching with 5 ml Alkaline Hypochlorite Solution (1 ml Sodium hypochlorite ~3%, 0.5 ml 5 M sodium hydroxide solution, and 3.5 ml water). Bleached eggs (F0 generation) were moved to the RNAi plates and hatched at ambient temperature prior to transfer and growth at 15°C. The F1 generation was allowed to reach late L4 stage prior to dissection and immunological analysis.

## Immunofluorescence

F1 adult worms were picked from RNAi/control plates and washed with 1 ml of 0.25 mM Levamisole (anthelmintic chemical) in wash buffer (1X phosphate-buffered saline with 1% (v/v) Tween20). Dissected gonads were transferred to a microcentrifuge tube and spun down at 8000 rpm for 2–3 s (all spinning steps were performed at this speed for 2–3 s). The supernatant was removed carefully, and extruded gonads were fixed with 200 µl 3% (v/v) formaldehyde for 30–60 min at room temperature. The microcentrifuge tubes were spun again, the supernatant was removed, and the gonads were then treated with 100% methanol and incubated at −20°C for at least 10 min. The microcentrifuge tubes were spun, supernatant was discarded and 100 µl of 10 mM sodium citrate buffer, that contained 0.05% (v/v) Tween20 adjusted to pH 6.0, was added for antigen retrieval. The gonads were incubated in sodium citrate buffer for 30 min, spun down and supernatant was removed. Following this, 100 µl of 3% (w/v) bovine serum albumin (BSA) in wash buffer was added to the microcentrifuge tubes for 30 min or longer to block non-specific staining, spun down and removed. Primary antibody, rabbit α-PHH3 (sc-8656-R Santa Cruz, 1:1000 dilution) was then added to the gonad pellet and incubated overnight at 4°C for efficient staining. The gonads were washed with 200 µl wash buffer three times for 30 min (10 min interval) and incubated with fluorescent Cy3-conjugated secondary α-Rabbit antibody (A10520 Invitrogen 1:2000 dilution), for 2 hr at room temperature. The gonads were washed with 200 µl wash buffer and incubated with 100 ng ml$^{-1}$ 4,6-diamidino-2-phenylindole hydrochloride (DAPI, D9542 Millipore Sigma) in PBS for 10 min to stain DNA. The gonads were washed three times for 30 min (10 min interval) and 10–20 µl supernatant was left behind in the microcentrifuge tube. Fluorescent images were generated using an Olympus FV3000RS confocal laser scanning microscope. Images were analyzed using FLUOVIEW FV3000 software.

## Immunoblots

Yeast cells were lysed by ultrasonication (Qsonica model number Q125, 25 Watts for 60 s with a 3 s interval after 3 s of sonication) in lysis buffer (50 mM Tris, pH 8, 500 mM NaCl, 1 mM EDTA pH 8.0, 5 mM DTT, 20 mM β-mercaptoethanol and 0.2% NP-40) containing protease inhibitors (Pierce protease inhibitor mini tablets, 1 tablet per 10 ml solution, ThermoScientific, cat. number A32955 and Phenylmethylsulfonyl flouride at a working concentration of 1 mM, Sigma-Aldrich, cat. number

88H0793). Clairified lysates was obtained by aspiration of the supernatant following by centrifugation at 14,000 × g for 20 min at 4°C. Samples were denatured by boiling in 2 × SDS page loading buffer (2x Laemmli Sample Buffer, Bio-Rad, cat. number 1610737) and separated on 10% SDS-PAGE gels before transferring to Immobilon-P membranes (Millipore). The membrane was blocked in 5% milk for 1 hr at room temperature prior to overnight incubation with HA antibody (1:1000 dilution; Anti-HA.11 epitope tag antibody, BioLegend, cat. number MMS-101R) overnight at 4°C. The secondary goat anti-mouse antibody conjugated to horseradish peroxidase (1:2000; Goat anti-mouse IgG (H + L), HRP conjugate, Proteintech, cat. number SA00001-1). Peroxidase activity was detected using Pierce ECL Western Blotting Substrate (Thermo Fisher) on ChemiDoc Touch Imaging System, Bio-Rad). The blot was stripped in Restore Plus western blot stripping buffer (Thermo Fisher, cat. number SL258473) according to the manufacturer's instructions and re-probed with a glyceraldehyde 3-phosphate dehydrogenase (GAPDH) antibody (1:2000 dilution; GAPDH antibody, Proteintech, cat. number 10494–1-AP) and goat anti-rabbit secondary antibody for GAPDH expression detection (1:10,000 dilution; Goat anti-rabbit IgG (H + L), HRP conjugate, Proteintech, cat. number SA00001-2).

## Acknowledgments

We are grateful to Jason Williams and the NIEHS Mass Spectrometry Research and Support Group for analyses that were crucial for the success of crystallization and Andrew Sikkema for advice and instruction on X-ray crystallography techniques. We thank John Gonczy for assistance with data collection at SER-CAT beamlines 22-ID and 22-BM at the Advanced Photon Source, Argonne National Laboratory, and Lars Pedersen and Juno Krahn for crystallographic and data collection support at NIEHS. We appreciate critical reading of this manuscript by our colleagues L Pedersen and M Wells. This work was supported in part by NIH grants R01NS100788 (ZTC), RGM122001A (TWL) and by the Intramural Research Program of the National Institutes of Health, National Institute of Environmental Health Sciences (TMTH). The Advanced Photon Source used for this study was supported by the US Department of Energy, Office of Science, Office of Basic Energy Sciences, under contract no. W-31–109-Eng-38.

## Additional information

### Funding

| Funder | Grant reference number | Author |
| --- | --- | --- |
| National Institutes of Health | RGM122001A | Te-Wen Lo |
| National Institute of Environmental Health Sciences | Intramural Research Program | Traci M Tanaka Hall |
| National Institutes of Health | R01NS100788 | Zachary T Campbell |

The funders had no role in study design, data collection and interpretation, or the decision to submit the work for publication.

### Author contributions

Vandita D Bhat, Zachary T Campbell, Supervision, Funding acquisition, Writing—original draft, Project administration, Writing—review and editing; Kathleen L McCann, Yeming Wang, Dallas R Fonseca, Tarjani Shukla, Jacqueline C Alexander, Chen Qiu, Marv Wickens, Investigation, Writing—original draft, Writing—review and editing; Te-Wen Lo, Writing—original draft, Project administration, Writing—review and editing; Traci M Tanaka Hall, Supervision, Writing—original draft, Project administration, Writing—review and editing

### Author ORCIDs

Kathleen L McCann (iD) https://orcid.org/0000-0002-7144-4851
Traci M Tanaka Hall (iD) https://orcid.org/0000-0001-6166-3009
Zachary T Campbell (iD) http://orcid.org/0000-0002-3768-6996

Decision letter and Author response
Decision letter https://doi.org/10.7554/eLife.43788.031
Author response https://doi.org/10.7554/eLife.43788.032

## Additional files

### Supplementary files
• Transparent reporting form
DOI: https://doi.org/10.7554/eLife.43788.021

### Data availability
All data associated with the manuscript are present in the source data file. Data have also been deposited to PDB under the accession numbers 6NOD, 6NOH, 6NOF, and 6NOC.

The following datasets were generated:

| Author(s) | Year | Dataset title | Dataset URL | Database and Identifier |
|---|---|---|---|---|
| Bhat VD, McCann KL, Wang Y, Fonseca DR, Shukla T, Alexander JC, Qiu C, Wickens M | 2019 | PUF-8 data from Engineering a conserved RNA regulatory protein repurposes its biological function in vivo | https://www.rcsb.org/structure/6NOD | Protein Data Bank, 6NOD |
| Bhat VD, McCann KL, Wang Y, Fonseca DR, Shukla T, Alexander JC, Qiu C, Wickens M | 2019 | SS/Y data from Engineering a conserved RNA regulatory protein repurposes its biological function in vivo | https://www.rcsb.org/structure/6NOH | Protein Data Bank, 6NOH |
| Bhat VD, McCann KL, Wang Y, Fonseca DR, Shukla T, Alexander JC, Qiu C, Wickens M, Lo T-W, Tanaka Hall TM, Campbell ZT | 2019 | AS/Y data from Engineering a conserved RNA regulatory protein repurposes its biological function in vivo | https://www.rcsb.org/structure/6NOF | Protein Data Bank, 6NOF |
| Bhat VD, McCann KL, Wang Y, Fonseca DR, Shukla T, Alexander JC, Qiu C, Wickens M, Lo T-W, Tanaka Hall TM, Campbell ZT | 2019 | AQ/Y data from Engineering a conserved RNA regulatory protein repurposes its biological function in vivo | https://www.rcsb.org/structure/6NOC | Protein Data Bank, 6NOC |

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
