## [Decision Letter]

Thank you for submitting your article "Engineering a conserved RNA regulatory protein repurposes its biological function in vivo" for consideration by *eLife*. Your article has been reviewed by two peer reviewers, and the evaluation has been overseen by Timothy Nilsen as Reviewing Editor and James Manley as the Senior Editor. The reviewers have opted to remain anonymous.

The reviewers have discussed the reviews with one another and the Reviewing Editor has drafted this decision to help you prepare a revised submission.

As you will see, both reviewers were positive about the work and thought that it was, in principle, suitable for *eLife*. Nevertheless both also raised a number of relatively minor points that should be addressed via revision.

*Reviewer #1:*

The manuscript by Bhat et al. describes an elegant structure-function analysis of two nematode members of the PUF domain RNA-binding proteins, PUF-8 and FBF-2. These two proteins have different functions, but are otherwise very similar. They recognize the same nucleotide sequence motif, but vary in the overall length of the motif recognized (8 vs. 9 nucleotides, respectively). The prevailing hypothesis is that the curvature of the PUF domain defines motif length. The primary finding here is that the identity of the amino acids in repeat 5 are the primary determinant of motif length, as opposed to curvature. This result matters to the community of scientists building designer PUF proteins, but also matters to those trying to understand how RNA-binding proteins accomplish specific recognition of mRNAs, and those interested in motif evolution.

There is much data in the manuscript to support the conclusions, including a structure of PUF-8, a modified yeast 3 hybrid assay that identified mutations of PUF-8 that alter its motif length preference, binding analysis of several PUF variants by gel shifts and 3-hybrid assays, more structures of mutant FBF-2 variants with differing motif binding activities, and a functional substitution experiment in worms demonstrating that modified FBF-2 can compensate for loss of PUF-8 to prevent germline tumors.

Overall, the work appears to be well executed and the data are reasonably interpreted. My primary issue with the manuscript is in the presentation, which could stand to be more succinct and have a clearer presentation of the logic behind the experiments (details below). It should be possible to improve the clarity of the manuscript without additional experiments. As such, I would support publication in *eLife* following revision.

1) The use of the term "genetic screen" in the Abstract is ambiguous. Be more precise, use "modified yeast 3-hybrid screen" or similar.

2) "R5 uses its TRM to directly contact the RNA base at position 4". Are you referring to R5 from FBF-2, PUM1, or PUF-8?

3) It is claimed that the modified yeast 3-hybrid approach increases the true positive rate, but there is no data to support the claim. This data should be shown, or the claim withdrawn.

4) Subsection "Changes in curvature are dispensable". I agree that the data show this to be the case, but the last paragraph of the Introduction and the Discussion both indicate that curvature and sequences contribute to motif length selection. This is confusing. I would encourage the authors to be clear about the relative contribution of curvature vs. amino acid identity throughout the manuscript.

5) Towards that end, please put numbers (i.e. fold change) on the magnitude of the effects.

6) I could not follow the rationale for the subsection "Compensatory mutations reveal engagement of the 3' end". This needs to be explained better. Why were these experiments done?

7) In the subsection “FBF-2 R5 TRM variants retain FBF-2 base recognition specificity at positions 3-5”, the data show that changes in R5 effect the binding specificity of adjacent repeats, which suggests cooperativity between repeats and violates the 1 repeat, 1 nucleotide design principle. The authors should point this out.

*Reviewer #2:*

The manuscript by Bhat and co-workers explores the molecular basis of target specificity by PUF proteins and defines the role that this specificity plays in protein function.

The authors use two proteins expressed in the germline of *C. elegans*, PUF-8 and FBF-2, to investigate how the PUM domain can recognise sequences of different length and, for the central positions, specificity. The data show that recognition of an RNA sequence of different length can be obtained by changing one-two amino acids in the central repeat of the PUM domain, and that the length recognised is not linked to the overall curvature of the RNA binding surface, at least in this case. Importantly, the authors use function recovery assays to show that altering the RNA recognition properties of the FBF-2 PUM domain so they resemble the ones of PUM domain of PUF-8 changes the function of FBF-2 to resemble the one of PUF-8. This shows that the functional differences of these two proteins are largely defined by the targets they recognise rather than by differences in the regulatory mechanism.

Overall, the paper is interesting, logical and well-written. The quality of the data is high and the figures are generally well drafted. I have a few comments, which are reported below.

1) The paper describes in detail the different structures in relation to RNA binding but could discuss in a more explicitly mechanistic fashion why specific mutations cause the changes in target specificity, e.g. why an arginine to tyrosine substitution changes the position of the base in position 5 or what is the origin of the changes in nucleobase preference.

2) In the last paragraph of the Introduction, the authors state that they provide evidence that both PUF protein curvature and TRM interactions account for the difference in binding length element of FBF-2 and PUF-8. My understanding of the data is that there is no evidence in this paper that a change in the curvature is required and the authors state this repeatedly later in the paper. This sentence should be change accordingly.

3) Figure 2A-C should probably be larger and the labels clearer.

4) Figure 1—figure supplement 1 and Figure 2—figure supplement 1 could be moved to the main paper.

---

## [Author Response]

Reviewer #1:[…] Overall, the work appears to be well executed and the data are reasonably interpreted. My primary issue with the manuscript is in the presentation, which could stand to be more succinct and have a clearer presentation of the logic behind the experiments (details below). It should be possible to improve the clarity of the manuscript without additional experiments. As such, I would support publication in eLife following revision.1) The use of the term "genetic screen" in the Abstract is ambiguous. Be more precise, use "modified yeast 3-hybrid screen" or similar.

The suggestion has been incorporated.

2) "R5 uses its TRM to directly contact the RNA base at position 4". Are you referring to R5 from FBF-2, PUM1, or PUF-8?

The sentence now indicates that we are referring to both PUF-8 and PUM1.

3) It is claimed that the modified yeast 3-hybrid approach increases the true positive rate, but there is no data to support the claim. This data should be shown, or the claim withdrawn.

The claim has been removed. The data will be presented in a methods oriented manuscript describing the new approach.

4) Subsection "Changes in curvature are dispensable". I agree that the data show this to be the case, but the last paragraph of the Introduction and the Discussion both indicate that curvature and sequences contribute to motif length selection. This is confusing. I would encourage the authors to be clear about the relative contribution of curvature vs. amino acid identity throughout the manuscript.

We agree that we did not present this clearly, and we have edited the manuscript to be more specific. We have revised paragraphs in the Introduction (fifth paragraph) and Discussion (second paragraph). We clarify that our PUF-8 structure provides new information that its curvature is distinct from FBF-2 and also that more information is needed to directly test the role of curvature in motif length specificity.

5) Towards that end, please put numbers (i.e. fold change) on the magnitude of the effects.

K_rel_ values have been added to Table 2.

6) I could not follow the rationale for the subsection "Compensatory mutations reveal engagement of the 3' end". This needs to be explained better. Why were these experiments done?

We have rewritten this section to improve the rationale behind the compensatory mutants. These experiments are critical to test the model implied by the crystal structure.

“Our crystal structures of the FBF-2 variants indicated that they used their TRMs to recognize the full 8-nt PRE. […] To test the 1:1 binding mode, we examined whether interactions of FBF-2 variants with the 3´ sequences of the 8-nt RNA are required for tight binding in cells.”

7) In the subsection “FBF-2 R5 TRM variants retain FBF-2 base recognition specificity at positions 3-5”, the data show that changes in R5 effect the binding specificity of adjacent repeats, which suggests cooperativity between repeats and violates the 1 repeat, 1 nucleotide design principle. The authors should point this out.

We thank the reviewer for mentioning this and have added the following text:

“This suggests that there is cooperativity between TRMs as opposed to true independent modularity. Similar results were obtained with variants at the repeat seven TRM (Campbell, Valley et al., 2014).”

Reviewer #2:[…] Overall, the paper is interesting, logical and well-written. The quality of the data is high and the figures are generally well drafted. I have a few comments, which are reported below.1) The paper describes in detail the different structures in relation to RNA binding but could discuss in a more explicitly mechanistic fashion why specific mutations cause the changes in target specificity, e.g. why an arginine to tyrosine substitution changes the position of the base in position 5 or what is the origin of the changes in nucleobase preference.

We have added more details about the interactions in the crystal structures (subsections “Changes in curvature are dispensable” and “The stacking residue is critical for binding length specificity”) with new Figures 3C, 4B, and 4C.

2) In the last paragraph of the Introduction the authors state that they provide evidence that both PUF protein curvature and TRM interactions account for the difference in binding length element of FBF-2 and PUF-8. My understanding of the data is that there is no evidence in this paper that a change in the curvature is required and the authors state this repeatedly later in the paper. This sentence should be change accordingly.

As noted for reviewer 1, point #4, we have tried to improve the presentation of what was known, what is now known, and what remains to be understood. We have revised paragraphs in the Introduction (fifth paragraph) and Discussion (second paragraph). We clarify that our PUF-8 structure provides new information that its curvature is distinct from FBF-2 and also that more information is needed to directly test the role of curvature in motif length specificity.

3) Figure 2A, B and C should probably be larger and the labels clearer.

This has been corrected.

4) Figure 1—figure supplement 1 and Figure 2—figure supplement 1 could be moved to the main paper.

We moved Figure 1—figure supplement 1 (PUF-8 structural figures) to the main paper. Unfortunately, we found it difficult to move Figure 2—figure supplement 1 (schematic of the yeast 3-hybrid experiment), because doing so would place figures out of order in the text.